# Development of DHODH inhibitors incorporating virtual screening, pharmacophore modeling, fragment-based optimization methods, ADMET, molecular docking, molecular dynamics, PCA analysis, and free energy landscape

**Qu Wang[1,2], Yu Hao Xu[1,2], Heng Jiang[3], Biao Deng[1], Zhu Liang [1]***

**1** Department of Thoracic Surgery, Affiliated Hospital of Guangdong Medical University, Zhanjiang, Guangdong, China, **2** Graduate School, Guangdong Medical University, Zhanjiang, Guangdong, China, **3** The First Clinical College, Guangdong Medical University, Zhanjiang, China

* liangzhuwsh@163.com

## Abstract

The overexpression of dihydroorotate dehydrogenase (DHODH) in various malignant tumor cells is significantly associated with ferroptosis, making DHODH inhibition a promising strategy for cancer therapy. In this study, we employed an integrated approach to screen and optimize DHODH inhibitor candidates. First, virtual screening of the FDA-approved drug library identified 20 potential compounds (with the positive control AG-636 as a benchmark, docking score: 133.166). Subsequent pharmacophore modeling (ROC curve value >0.8) further narrowed the candidates to six compounds, which underwent fragment displacement optimization. All optimized compounds were evaluated for absorption, distribution, metabolism, excretion, and toxicity (ADMET) properties. Molecular docking identified compounds 65:[(4S)-2,2-dimethyl-1,3-dioxolan-4-yl]methyl 3-(4-{[(2S)-2-hydroxypropyl]oxy} phenyl) (docking score: 197.362) and 66: [(4S)-2,2-dimethyl-1,3-dioxolan-4-yl]methyl 4-(4-{[(2S)-2-hydroxypropyl]oxy}phenyl) (docking score: 202.623) as high-affinity candidates. Molecular dynamics (MD) simulations, principal component analysis (PCA), and free energy landscape (FEL) analyses confirmed stable binding conformations for both compounds. Notably, compound 66: [(4S)-2,2-dimethyl-1,3-dioxolan-4-yl] methyl 4-(4-{[(2S)-2-hydroxypropyl]oxy}phenyl) exhibited minimal conformational changes, suggesting superior binding stability. This study advances compound 66: [(4S)-2,2-dimethyl-1,3-dioxolan-4-yl]methyl 4-(4-{[(2S)-2-hydroxypropyl]oxy}phenyl) as a promising DHODH inhibitor candidate through a multimodal workflow integrating structure-based pharmacophore design, fragment optimization, ADMET profiling, and advanced molecular simulations, providing a novel avenue for DHODH-targeted antitumor therapies.

**Data availability statement:** All relevant data are within the manuscript and its Supporting information files.

**Funding:** This project was supported by Zhanjiang Science and Technology Project (2025A502006, 2025B01261); Clinical Research Program, Affiliated Hospital of Guangdong Medical University (LCYJ2021A004, LCYJ2022DL003); National College Students' innovation and entrepreneurship training program (202510571006, 202410571016); Project funded by the School Planning, Construction and Development Center of the Ministry of Education (CSDP25LF8C439). All funding was received by author ZL.

**Competing interests:** The authors have declared that no competing interests exist.

**Abbreviations:** DHODH, Dihydroorotate dehydrogenase; ADMET, absorption, distribution, metabolism, excretion, and toxicity; MD, molecular dynamics; PCA, Principal Component Analysis; FEL, Free Energy Landscape.

## 1. Introduction

The Dihydroorotate dehydrogenase (DHODH), an essential mitochondrial enzyme, plays a pivotal role in the de novo synthesis of pyrimidines [1,2]. It catalyzes the conversion of dihydroorotate (DHO) to orotate (ORO), utilizing flavin mononucleotide (FMN) as a cofactor. This enzymatic reaction is fundamental for cellular DNA and RNA synthesis [3,4]. DHODH is frequently overexpressed in various malignancies, including acute myeloid leukemia (AML) [5,6], breast cancer [7,8], colorectal cancer (CRC) [9,10] and melanoma [11]. The heightened sensitivity of these cancer cells to DHODH inhibitors indicates the enzyme's potential as a therapeutic target [12]. The primary mechanism of DHODH inhibitors is the inhibition of pyrimidine synthesis, thereby curtailing the proliferation of cancer cells, which have an increased demand for pyrimidines due to their accelerated cell division and growth [3,13,14].

Recent research has highlighted the role of dihydroorotate dehydrogenase (DHODH) in countering ferroptosis, a form of regulated cell death attributed to lipid peroxide accumulation. DHODH's functionality extends beyond pyrimidine synthesis to include mitigation of ferroptosis in mitochondria by modulating dihydro ubiquinone production, thereby resisting cell death [15]. DHODH inhibitors have emerged as a therapeutic strategy for cancer by promoting ferroptosis by inhibiting ferroptosis suppressor protein-1 (FSP1), independent of DHODH's catalytic activity [16]. This suggests that DHODH inhibitors could enhance cancer cell demise by modulating mitochondrial coenzyme Q10 (CoQ10) to prevent lipid peroxidation. The interaction between DHODH and glutathione peroxidase 4 (GPX4), a key ferroptosis defense, is significant; cells with low GPX4 expression are more susceptible to ferroptosis upon DHODH inhibition [17–19]. Clinical trials with the DHODH inhibitor Brequinar have shown efficacy in inducing ferroptosis and suppressing tumor growth in cancers with low GPX4 expression [20]. The structural basis of DHODH, featuring an α/β barrel domain and an α-helical domain, is crucial for the design of specific inhibitors that bind within a tunnel connecting these domain [21–23].

Numerous dihydroorotate dehydrogenase inhibitors (DHODH) have advanced to clinical trials to treat various oncological conditions [24]. Leflunomide, initially used for rheumatoid arthritis, has demonstrated anti-neoplastic effects in both in vitro studies and animal models [25,26]. However, its clinical application is limited by adverse effects. Brequinar, a potent DHODH inhibitor, showed early promise in clinical trials for its antitumor activity. Yet, its further clinical development was discontinued due to insufficient therapeutic efficacy [27].

Computer-aided drug design (CADD) can rapidly screen various compounds to identify potential drug candidates. In previous studies, we have successfully screened many inhibitors against tumor targets from the FDA-approved drug database and the Marine Compound Database [28–30]. Pharmacophore screening identifies common features among known molecular compounds [31,32]. Fragment-based drug design optimizes and replaces molecular fragments from existing libraries to create new compounds with enhanced properties, ultimately refining the docking space [33–35].

This research identified six promising lead compounds through virtual screening and pharmacophore validation of an FDA-approved drug library, focusing on structural characteristics. These compounds were subsequently compared against the previously reported AG636. We selected two lead compounds that exhibited a strong binding affinity for DHODH, following fragment replacement modification and ADMET property optimization. Molecular docking, molecular dynamics simulations, PCA analysis and FEL indicated that compounds 66: [(4S)-2,2-dimethyl-1,3-dioxolan-4-yl]methyl 4-(4-{[(2S)-2-hydroxypropyl]oxy}phenyl) possess significant potential for in vivo application. Our methodology is depicted in Fig 1.

## 2. Materials and methods

### 2.1 Database construction and molecular preparation

From the FDA's drug Compounds database, we selected 6,140 molecules for virtual screening. The X-ray structure of DHODH complexed with the 2-Hydroxypyrazolo[1,5-a] pyridine inhibitor, MEDS433, was chosen as the structural model for DHODH (PDB code: 6FMD) [34,36,37]. Additionally, the docking active sites were determined concerning the study previously published by Galati S et al [38]. Utilizing the Discovery Studio platform version 2019, the Libdock module facilitated rapid high-throughput screening. This process involved the random generation of 100 hotspots within a spatial sphere defined by the active residues. Concurrently, each compound was also generated in 100 conformations during preliminary preparation, with these conformations aligning with the hotspots during the docking process. Ultimately, the

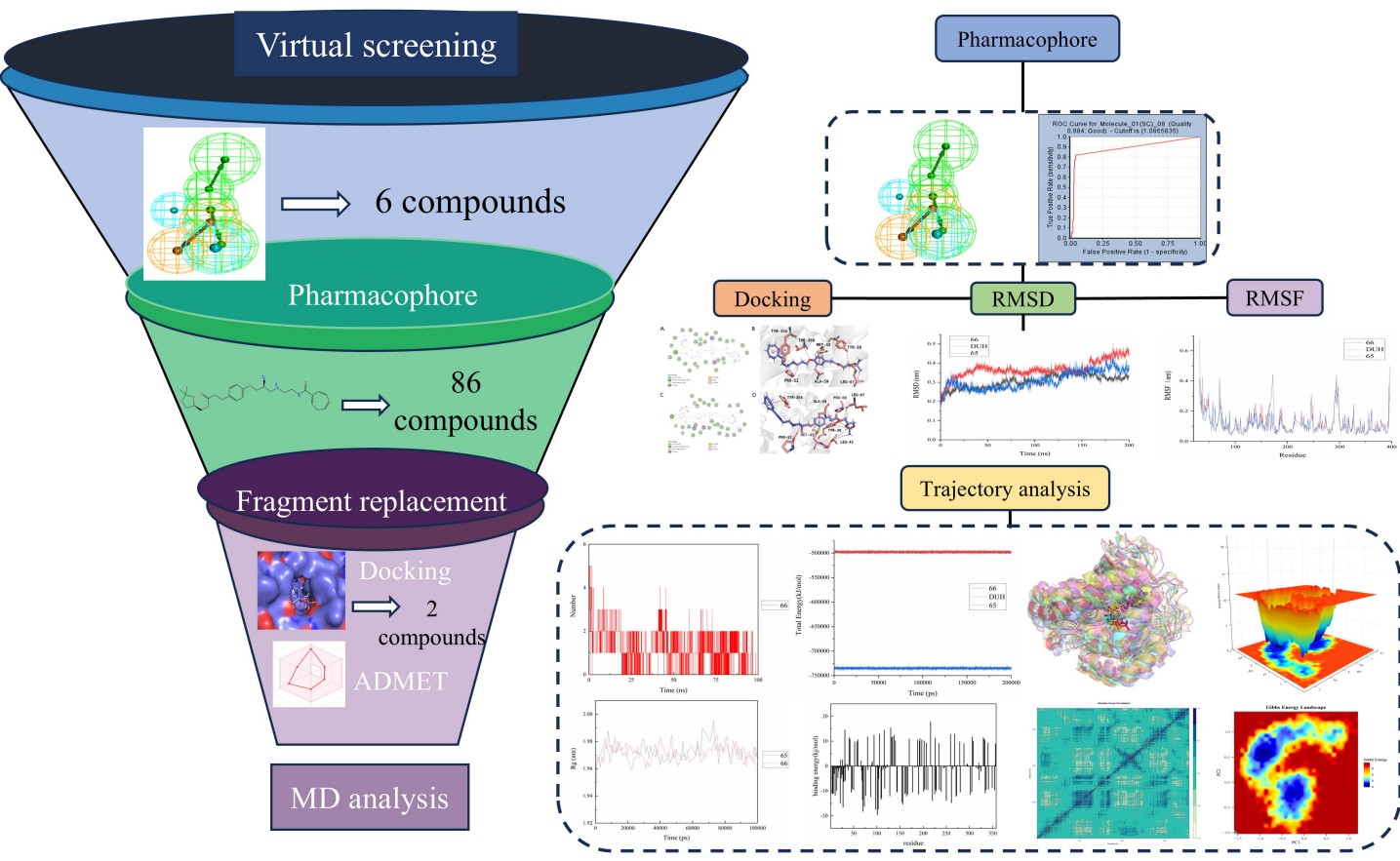

**Fig 1. Workflow of this study.**

Libdock module assigned scores reflective of the alignment degree, correlating to the strength of the ligand-receptor interaction.

## 2.2 Pharmacophore construction

We have amassed a collection of 69 DHODH inhibitors, sourced from both the scientific literature and the ChEMBL database (https://www.ebi.ac.uk/chembl/), for our study [39]. From this collection, 30 structurally similar small molecules were carefully selected to generate a pharmacophore model, as detailed in S1 Table. The remaining inhibitors served as a validation set of active small molecules, listed in S2 Table. Furthermore, we subjected these 30 small molecules to the DUD-E platform (https://dude.docking.org/), where they formed a bait set of 300 compounds, used to validate the activity of the small molecules. Employing Discovery Studio 2019, pharmacophore models were created for the aforementioned 30 small molecules based on their shared molecular characteristics. The Common Feature Pharmacophore Generation tool was utilized, with Conformation Generation set to the 'BEST' option and Maximum Conformations expanded to 200. We configured the settings to discard existing conformations, with 'Discard Existing Conformations' enabled, an 'Energy Threshold' of 10, and disabled options for saving conformations and parallel processing. Once the settings were finalized, we specified the features as HB_ACCEPTOR, HB_DONOR, HYDROPHOBIC, POS_IONIZABLE, NEG_IONIZABLE, and RING_AROMATIC. The 'Maximum Pharmacophores' were limited to 10, with a 'Minimum Interfeature Distance' of 2.97 Å and 'Maximum Excluded Volumes' set to 0. With validation enabled, we employed the 30 DHODH inhibitors as Active Ligands and complemented them with an additional 300 decoy set molecules as Inactive Ligands, culminating in the generation of 10 pharmacophore models. An important branch of machine learning is Bayesian classification model. The Bayesian classification model is constructed using 30 active molecules that generate pharmacophors and 363 active molecules collected by chembl website. According to their structure and physical and chemical properties, a total of 300 decoy molecules of the DUD-E website and 239 molecules collected by chembl website are generated as inactive molecules. On the Discovery Studio platform, the key molecular descriptors obtained through dimensionality reduction in PCA and molecular fingerprints such as ECFP_n are applied to the proposed model construction process. Various fingerprint-descriptor combinations were tried during the model construction process. The area under the ROC curve (AUC) of the model is taken as an indicator of classification ability, and the best Bayesian model is selected for further analysis, in which active labels are used for classification.

## 2.3 Fragment replacement of the lead compounds

Fragment-based drug discovery (FBDD) represents a strategic approach to the development of potent lead molecules. Among the compound molecules obtained from the initial screening, the fragments to be replaced are selected based on the following criteria: structural units with weak non-covalent interactions (such as hydrogen bonds, hydrophobic interactions, electrostatic interactions, etc.) with the target protein; Groups with potential toxicity risks (such as aromatic rings and other fragments that may cause biocompatibility issues). For the above-mentioned screened fragments, structural replacement is required, and the replaced molecules must comply with the five rules of Lipinski drugs to ensure their good oral bioavailability and metabolic stability. In Discovery Studio 2019, the fragment replacement protocol facilitates the systematic optimization of compounds. Initially, compounds undergo energy minimization using the CHARMM force field, ensuring structural stability. Subsequently, the software executes fragment replacement calculations utilizing its default library, which comprises 1,495,478 fragments. Fragment similarity is evaluated based on the presence of cyclic/aromatic rings and molecular surface area, thereby pinpointing fragments that closely resemble the originals in physicochemical properties. These fragments are then subjected to further molecular structure optimization to augment their interaction potential with proteins. The final step involves Pareto ranking, which considers fragment-protein interactions, adherence to Lipinski's rules, receptor bump matching, and the novelty of the fragment, encompassing chain combinations, double bonds, aromatic bonds, and the distribution of nitrogen (N), sulfur (S), and oxygen (O) atoms. This comprehensive

process identifies top-performing molecules and selects strong candidates for drug design. Ultimately, the molecules are re-docked to the proteins using LibDock, with those exhibiting higher docking scores than the original compounds being selected for subsequent screening stages.

### 2.4 Absorption, Distribution, Metabolism, Excretion, and Toxicity (ADMET)

Assessment of pharmacokinetics and toxicity (ADMET) is instrumental in the process of screening and winnowing out molecules with suboptimal drug-like properties from a vast array of candidates. Utilizing Discovery Studio's Calculate Molecule Properties tool, we evaluated the ADME profiles of 86 compounds [40–42]. We predicted four key pharmacokinetic parameters: intestinal absorption (HIA), potential for hepatotoxicity, inhibitory activity against cytochrome CYP2D6, and aqueous solubility at 25°C. CYP2D6 is crucial for the metabolic breakdown of drugs within the body. Optimal intestinal absorption and minimal CYP2D6 inhibition are desirable as they can maximize the duration of drug action in the human body. Compounds exhibiting poor water solubility, inadequate intestinal absorption, or significant CYP2D6 inhibition were excluded from further consideration. The remaining compounds, which met the criteria, were advanced for additional analysis.

### 2.5 Molecular docking

The molecular docking workflow was systematically executed in two sequential phases. In the initial screening phase, rapid virtual screening of two pre-selected candidate compounds (identified through ADMET profiling) against dihydroorotate dehydrogenase (DHODH) was performed. Prior to docking simulations, structural preprocessing was conducted using PyMOL2.3.2 to optimize the receptor structure: water molecules, co-crystallized ligands, and non-essential metal ions were removed while retaining the essential flavin mononucleotide (FMN) cofactor. Then the processed receptor structure was saved in PDB format for subsequent docking calculations. The binding pocket of DHODH was defined by the original ligand's binding conformation, with the geometric centroid coordinates at (31.583Å, 11.130Å, 1.999Å). This active site encompasses 19 critical residues: Tyr-38, Leu-42, Met-43, Leu-46, Gln-47, Ala-59, Phe-62, Leu-67, Leu-68, Pro-69, Phe-98, Met-111, Val-134, Arg-136, Val-143, Tyr-356, Leu-359, Thr-360, and Pro-364, which are essential for ligand recognition and binding [43]. The prepared structures and docking sites were used for molecular docking in Autodock vina1.2.4. Subsequently, the CDOCKER program was employed to dock the two molecules, which exhibited favorable Libdock scores, with DHODH, facilitating an in-depth analysis of their interactions with the target. The Pose Cluster Radius was configured to 0.1, and the ligands were set to randomly generate 10 conformations, thereby enabling semi-flexible docking between the ligand and receptor.

### 2.6 Molecular dynamics

In this research, we determined the dynamic behavior of complexes 65:[(4S)-2,2-dimethyl-1,3-dioxolan-4-yl]methyl 3-(4-{[(2S)-2-hydroxypropyl]oxy}phenyl) and 66: [(4S)-2,2-dimethyl-1,3-dioxolan-4-yl]methyl 4-(4-{[(2S)-2-hydroxypropyl]oxy}phenyl) and DUH (crystal inhibitor) over a 200 ns simulation period, focusing on the conformational changes influenced by solvent exposure and periodic boundary conditions. We assessed the proteins' potential as dynamic targets and the interaction profiles of all candidate compounds. The simulation commenced with the construction and export of PDB files for the two molecules and the receptor protein using the Discovery Studio platform. The ligand, DUH and FMN topology was generated utilizing the GAFF force field via the ACPYPE online server (https://www.bio2byte.be/acpype/) [44]. Concurrently, the protein topology file was prepared using GROMACS version 2019, adopting the MD AMBER99SB-ILDN force field and the TIP3P water model [45,46]. A cubic simulation box, measuring 1.2 nm in radius, was established to encapsulate the protein-receptor complex, which was solvated with the SPC/E water model to mimic the aqueous environment. The system's charge neutrality was ensured by incorporating equivalent amounts of sodium and chloride ions. Following the successful assembly of the simulation system, energy minimization was conducted over 50,000 steps

at a temperature of 300 K. Subsequently, the receptor, ligand, and solvent were equilibrated under constant temperature and volume (NVT) and constant temperature and pressure (NPT) conditions for 25 ps, with a step size of 25,000 steps. A 100 ns molecular dynamics (MD) simulation was then executed. Ultimately, we collected and analyzed data on root mean square deviation (RMSD), root mean square fluctuation (RMSF), sol-vent-accessible surface area (SASA), the radius of gyration (Rg), total potential energy, and the number of hydrogen bonds [47].

## 2.7 MM-PBSA

MM/PBSA method is widely used in the calculation of free energy of receptor-ligand binding. This method is called Molecular Mechanics/Poisson Boltzmann (Generalized Born) Surface Area). The basic principle is to calculate the difference between the bound and unbound free energies of two solvated molecules or to compare the free energies of different solvated conformations of the same molecule. We extracted a stable 10 ns from the trajectory for calculation. The following Equation below describes the binding free energy and the resulting output formula is related to the calculated energies of the ligand and receptor [48].

$$G_{binding} = G_{protein} - (G_{complex} + G_{ligand})$$

(1)

The free energy of the protein-ligand complex is expressed by $G_{Complex}$, $G_{protein}$ represents the free energy of the protein in the solvent, and $G_{ligand}$ represents the free energy of the ligand in the solvent.

## 2.8 Principal component analysis based on free energy landscape and structural analysis

A principal component analysis (PCA) was performed on the MD simulation trajectories of complexes 65:[(4S)-2,2-dimethyl-1,3-dioxolan-4-yl]methyl 3-(4-{[(2S)-2-hydroxypropyl]oxy}phenyl) and 66: [(4S)-2,2-dimethyl-1,3-dioxolan-4-yl] methyl 4-(4-{[(2S)-2-hydroxypropyl]oxy}phenyl) and DUH (crystal inhibitor). This analysis focused on the most relevant dynamic interactions between proteins and inhibitors. We concentrate on the backbone atoms of the system to emphasize the key motions that control ligand binding. Principal component (PC) were obtained from the trajectory data using gmx covar and gmx anaeig in GROMACS [49–51]. This process required narrowing down the multidimensional dataset to identify the underlying axes of motion PCA shows a large number of conformational changes and patterns during protein-ligand interactions, providing insight into the dynamic characterization of binding. After generating the file pc12_gibbs~. xyz, the R language library Bio3D was used to generate principal component analysis result plots for the trajectories [52]. The corresponding PCs then serve as the coordinates for the creation of the free energy maps. Protein conformational changes associated with different energy states can be effectively revealed using free energy maps (FEL) [5,53]. These topographical maps show the distribution of structural states, where valleys indicate stable low-energy conformations. The purpose of studying free energy landscapes is to determine the stability and flexibility of the complex system Key observations include the identification of stable conformational substrates and transition states, which indicate the binding affinity and specificity of the inhibitor. This method enables us to find energetically favorable binding modes and predict the possible efficacy of the inhibitor. With the help of xpm2all.bsh script the data was processed and an energy landscape view was generated and finally the trajectories were extracted at each energy minimization stage after using VMD processed xtc trajectory files and the extracted structures were superimposed and visualized in pymol 2.3.0.

## 3. Results

### 3.1 Virtual screening

Virtual screening is frequently utilized to identify compounds within extensive libraries likely to engage with specific biological targets, potentially offering therapeutic or biological significance. In this study, 6,140 compounds from the FDA compound database were subjected to virtual screening using Discovery Studio's Libdock module and Autodock vina,

targeting the DHODH protein model (PDB code: 6FMD) obtained from the Protein Data Bank (PDB). AG-636, a well-documented DHODH inhibitor with an IC50 value of 17 nM, served as a positive control to enhance the accuracy of the docking process. The screening yielded 110,345 feasible conformations with docking scores varying from 18.5304 to 206.88, whereas the Libdock score for AG-636 was 133.166. Generally, higher Libdock scores suggest superior binding affinity. Consequently, the top 20 compounds, as indicated by their Libdock scores, were advanced to the next screening stage, thereby narrowing down the candidate pool. In the Autodock Vina program, the lower the score, the tighter the binding. It can also be seen that the vina scores of the screened compounds are all lower than those of the control compound AG-636. In addition, the pKa values are all less than 8 (Table 1).

### 3.2 Pharmacophore

Pharmacophore models facilitate the simulation of ligand molecules' active conformations by employing conformational searching and molecular superposition techniques. These models are instrumental in deducing and elucidating potential interaction patterns between receptor and ligand molecules. In this study, Discovery Studio 2019 was utilized to develop pharmacophore models grounded in the shared features of molecules, resulting in the generation of a total of 10 pharmacophores. Following validating a pharmacophore model, as detailed in Table 2, we identified a pharmacophore model CF_8, characterized by at least four features and exhibiting a Sensitivity, Specificity, and ROC Curve value exceeding 0.8, for further investigation. Fig 2 illustrates the pharmacophore model CF_8. In the preceding screening phase, 20 potential candidate compounds were identified. These 20 small molecules underwent virtual screening based on the CF_8 pharmacophore, leading to the selection of 6 small molecules for subsequent molecular fragment replacement optimization.

**Table 1. Libdock and Autodock vina、pka scores of 20 selected molecules and positive control AG-636 with DHODH. In addition, the DUH (crystal inhibitor) also performed docking and was shown in S3 Fig. It can be seen that two hydrogen bond interactions were formed, and the final docking score was −15.6344.**

| Molecules | Libdock score | Autodock vina | pka | Formula |
|---|---|---|---|---|
| Molecule85 | 192.849 | −7.1 | 6.54 | $C_{20}H_{23}CaN_7O_6$ |
| Molecule137 | 191.758 | −7.0 | 6.33 | $C_{20}H_{23}CaN_7O_6$ |
| Molecule639 | 188.329 | −6.9 | 6.71 | $C_{22}H_{28}N_4O_6$ |
| Molecule642 | 189.542 | −6.9 | 6.66 | $C_{22}H_{28}N_4O_6$ |
| Molecule643 | 188.688 | −6.8 | 6.55 | $C_{22}H_{28}N_4O_6$ |
| Molecule1106 | 194.315 | −7.3 | 6.38 | $C_{58}H_{73}N_7O_{17}$ |
| Molecule1281 | 196.975 | −7.3 | 6.78 | $C_{25}H_{40}ClN_3O_8$ |
| Molecule1282 | 206.88 | −7.8 | 6.24 | $C_{25}H_{40}ClN_3O_8$ |
| Molecule1768 | 203.663 | −7.7 | 6.66 | $C_{36}H_{55}NO_{15}$ |
| Molecule1769 | 197.529 | −7.5 | 6.78 | $C_{36}H_{55}NO_{15}$ |
| Molecule1786 | 197.096 | −7.4 | 6.92 | $C_{36}H_{64}Cl_2N_4$ |
| Molecule1792 | 190.861 | −7.5 | 6.35 | $C_{25}H_{39}NO_3$ |
| Molecule2929 | 188.42 | −7.1 | 6.71 | $C_{25}H_{22}O_{10}$ |
| Molecule4410 | 188.964 | −7.0 | 6.81 | $C_{35}H_{38}Cl_2N_8O_4$ |
| Molecule6043 | 200.143 | −7.8 | 6.95 | $C_{23}H_{28}F_2N_6O_4S$ |
| Molecule6142 | 188.54 | −6.9 | 6.31 | $C_{20}H_{31}CaN_7O_{12}$ |
| Molecule6164 | 192.827 | −7.2 | 6.21 | $C_{20}H_{31}CaN_7O_{12}$ |
| Molecule6443 | 189.878 | −7.1 | 6.32 | $C_{19}H_{19}N_7O_6$ |
| Molecule6444 | 192.407 | −7.4 | 6.21 | $C_{19}H_{19}N_7O_6$ |
| Molecule6454 | 190.297 | −7.0 | 6.22 | $C_{19}H_{19}N_7O_6$ |
| AG-636 | 133.166 | −6.0 | 6.34 | $C_{21}H_{17}N_3O_2$ |

**Table 2. The number of features, feature set, sensitivity, specificity, and ROC curve of 10 pharmacophore models based on the common characteristics of molecules.**

| Pharmacophore | Features | Sensitivity | Specificity | True Negatives | False Positives | ROC Curve |
|---|---|---|---|---|---|---|
| CF_1 | RHHAA | 0.74359 | 0.97333 | 5 | 0 | 0.866 |
| CF_2 | RHHAA | 0.74359 | 0.98333 | 5 | 0 | 0.865 |
| CF_3 | HHHAA | 0.69231 | 0.99333 | 5 | 0 | 0.843 |
| CF_4 | RHHAA | 0.74359 | 0.97333 | 5 | 0 | 0.863 |
| CF_5 | RHHAA | 0.79487 | 0.97333 | 5 | 0 | 0.881 |
| CF_6 | RHHAA | 0.76923 | 0.95667 | 5 | 0 | 0.875 |
| CF_7 | RHHAA | 0.79487 | 0.96667 | 5 | 0 | 0.883 |
| CF_8 | RHHAA | 0.82051 | 0.95667 | 5 | 0 | 0.884 |
| CF_9 | HHHAA | 0.64103 | 0.99000 | 5 | 0 | 0.815 |
| CF_10 | RHHAA | 0.79487 | 0.98000 | 5 | 0 | 0.882 |

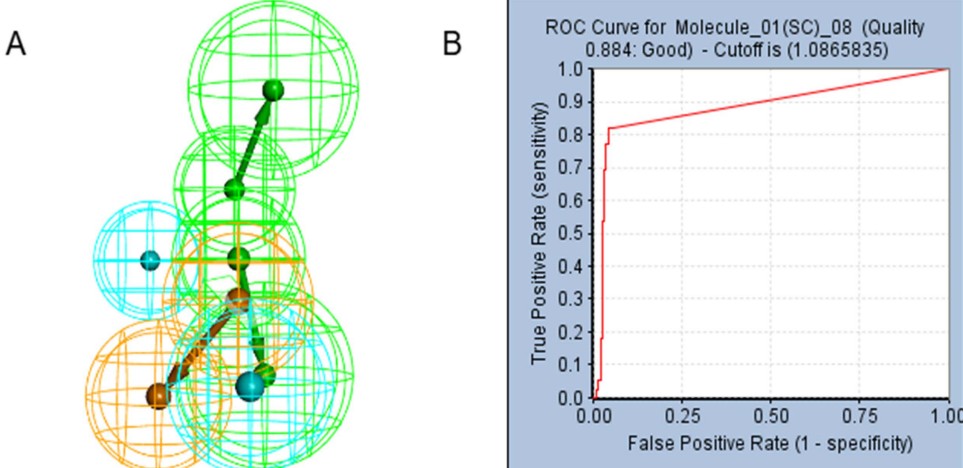

**Fig 2. The pharmacophore model and Receiver operating characteristic (ROC) curve validation. (A)** Pharmacophore model CF_8. Hydrophobic group features are represented by blue spheres, hydrogen bond donor features are represented by green spheres, and aromatic ring features are represented by orange spheres **(B)** ROC curve.

Where active molecules are marked as active (i.e., yes) and inactive molecules as inactive (i.e., no). When the absolute value of the Bayesian score is greater than the critical value, the compound is classified as "active"; otherwise, it is classified as "inactive". Its ROC curve is shown in S1 and S2 Figs. The Naive Bayes classification model demonstrates excellent recall ability. The Naive Bayes (NB) model demonstrates excellent classification and recognition capabilities. For the classification of active molecules, the AUC of the model for predicting true positives is 0.993. For inactive molecules, the AUC of the model for true positivity prediction is 0.994, close to 1, which indicates its excellent classification ability.

### 3.3 Optimization of fragment replacement of candidate lead molecules

By screening small molecular fragments with weak affinities for target proteins and subsequently optimizing their structures, we can enhance the activity of lead compounds. We conducted fragment substitution for six small molecules previously identified in our screening process (Table 3). We assessed their interactions with the protein and prioritized the replacement of fragments with minimal protein interactions. Following the fragment replacement, compound 4410 yielded

**Table 3. Structure, Libdock of the positive control AG636 and six lead molecules.**

| Molecule | Libdock score | CDOCKER ENERGY | CDOCKER INTERACTION ENERGY |
|----------|---------------|----------------|----------------------------|
| AG636 | 133.166 | −17.6145 | −56.6193 |
| 4410 | 188.964 | −20.8563 | −62.3414 |
| 1792 | 190.861 | −21.4878 | −61.4575 |
| 6043 | 200.143 | −22.4789 | −63.782 |
| 1281 | 196.975 | −47.6895 | −66.1551 |
| 1282 | 206.88 | −21.7741 | −64.2174 |
| 1106 | 194.315 | −21.5789 | −64.3387 |

97 new small molecules, compound 1281 yielded 66: [(4S)-2,2-dimethyl-1,3-dioxolan-4-yl]methyl 4-(4-{[(2S)-2-hydroxypropyl]oxy}phenyl), compound 1282 yielded 70, and compound 1106 yielded 82 new small molecules. However, upon further analysis, compound 4410 produced 73 new small molecules, compound 1281 produced 5, compound 1282 produced 7, and compound 1106 produced 1 new small molecule. Using Discovery Studio 2019 (DS2019), we performed Libdock to select molecules with a higher docking score than their parent compounds. Compounds 1792 and 8043 did not exhibit improved scores over the originals; their docking scores were notably lower than those of the new small molecules generated by fragment replacement. Consequently, these two initial compounds were excluded from further consideration. The remaining small molecules underwent ADMET analysis.

### 3.4 Absorption, Distribution, Metabolism, Excretion, and Toxicity (ADMET) analysis

ADMET (Absorption, Distribution, Metabolism, Excretion, and Toxicity) profiling is a pivotal aspect of drug screening and design, encompassing the drug's absorption, distribution, metabolism, excretion, and potential toxicity within the body. These properties are crucial for determining drug candidates' safety, efficacy, and pharmacokinetics. We conducted an ADMET analysis on 86 molecules post-fragment substitution using the Calculate Molecular Properties module within the Discovery Studio platform. Key predictions included blood-brain barrier (BBB) permeability, aqueous solubility, intestinal absorption, hepatotoxic potential, and CYP2D6 enzyme inhibition. These properties were quantified and ranked categorically. We excluded all compounds exhibiting hepatotoxicity or CYP2D6 inhibition to identify molecules with favorable drug-like properties. The aqueous solubility, intestinal absorption, hepatotoxic potential, and CYP2D6 inhibition data for the remaining seven molecules are presented in Table 3. The Absorption Level, denoting good (0) to average (1) absorption, suggested that 20 compounds had moderate solubility and absorption rates. These compounds also showed low hepatotoxicity and cytochrome enzyme inhibition. Solubility values represent the compounds' aqueous solubility at 25°C; a Solubility Level of 4 indicates high solubility, enhancing dissolution in the gastrointestinal tract and promoting absorption and pharmaceutical efficacy. The blood-brain barrier (BBB) primarily shields the brain from harmful blood-borne substances while preserving cerebral environment stability. A BBB Level 4 indicates an unreliable prediction regarding permeability. Our final ADMET predictions identified these seven compounds as the most promising among the 86 assessed, highlighting their non-hepatotoxic nature, high aqueous solubility, and overall drug-like characteristics (Table 4).

### 3.5 Molecular docking

Based on the screening outcomes, we identified two promising compounds (Table 5): 1281_65:[(4S)-2,2-dimethyl-1,3-dioxolan-4-yl]methyl 3-(4-{[(2S)-2-hydroxypropyl]oxy}phenyl) and 1281_66: [(4S)-2,2-dimethyl-1,3-dioxolan-4-yl]methyl 4-(4-{[(2S)-2-hydroxypropyl]oxy}phenyl), both originating from compound 1281. These compounds demonstrated enhanced scores in both the Libdock and CDOCKER docking assessments, along with favorable ADMET toxicity profiles. Salvatore Galati et al. previously utilized the Hydroxypyrazo-lo[1,5-a]pyridine inhibitor MEDS433 for docking with the DHODH protein

**Table 4. ADME prediction results of 7 lead compounds.**

| Molecule | Absorption Level | CYP2D6 Prediction | EXT PPB Prediction | EXT Hepatotoxic Prediction | Solubility Level | BBB Level |
|----------|-----------------|-------------------|--------------------|----------------------------|-----------------|-----------|
| 1281_32 | 0 | false | false | false | 4 | 4 |
| 1281_65 | 0 | false | false | false | 4 | 3 |
| 1281_66 | 0 | false | false | false | 4 | 3 |
| 1281_64 | 1 | false | false | false | 4 | 4 |
| 1282_2 | 1 | false | false | false | 4 | 4 |
| 1282_43 | 1 | false | false | false | 4 | 4 |
| 1282_51 | 1 | false | false | false | 4 | 4 |

**Table 5. Structure, Libdock, and CDOCKER energy scores of original compound 1281 and its derivatives 1281_65 and 1281_66.**

| Molecule | Libdock score | CDOCKER ENERGY | CDOCKER INTER-ACTION ENERGY |
|----------|--------------|----------------|------------------------------|
| AG-636 | 133.166 | −17.6145 | −56.6193 |
| 1281 | 196.975 | −47.6895 | −66.1551 |
| 1281_65:[(4S)-2,2-dimethyl-1,3-dioxolan-4-yl]methyl 3-(4-{[(2S)-2-hydroxypropyl]oxy}phenyl) | 197.362 | −16.824 | −67.6248 |
| 1281_66:[(4S)-2,2-dimethyl-1,3-dioxolan-4-yl]methyl 4-(4-{[(2S)-2-hydroxypropyl]oxy}phenyl) | 202.623 | −27.6725 | −67.3524 |

model (PDB code: 6FMD). Our study also considered the protein residues highlighted in their research, including Tyr-38, Leu-42, Met-43, Leu-46, Gln-47, Ala-59, Phe-62, Leu-67, Leu-68, Pro-69, Phe-98, Met-111, Val-134, Arg-136, Val-143, Tyr-356, Leu-359, Thr-360, and Pro-364, to pinpoint the molecular docking sites. As depicted in Fig 3A and 3B, the positive control AG636 exhibited extensive interactions with the protein, engaging with the majority of these critical residues. This robust interaction is likely due to the structural resemblance between the positive control and MEDS433, both featuring benzene rings. Conversely, the original compound 1281 (Fig 3C and 3D) showed diminished interaction with the protein, potentially because of its more compact structure and fewer benzene rings. Nevertheless, the fragment substitution yielded satisfactory results. Compared to the original compound, the novel compound 1281_65:[(4S)-2,2-dimethyl-1,3-dioxolan-4-yl] methyl 3-(4-{[(2S)-2-hydroxypropyl]oxy}phenyl) (Fig 4A and 4B) established a more stable hydrogen bond with the protein residue Tyr-356 on the substituted fragment, a crucial site on the protein. Additional interactions with other key residues were also observed, likely a result of conformational changes post-fragment substitution, enhancing the compound's accessibility to further key residues. Similarly, the new compound 1281_66: [(4S)-2,2-dimethyl-1,3-dioxolan-4-yl]methyl 4-(4-{[(2S)-2-hydroxypropyl]oxy}phenyl) (Fig 4C and 4D) consistently interacted with key residues. Both novel compounds share the substituted moiety, cyclohexyl-1,3,5-triene, but vary in their attachment points to the unsubstituted portion, leading to subtle differences in their interactions. However, both compounds effectively engage with key residues on this ring. The structure of the positive control also features multiple benzene rings, implying that compounds with such ring structures may exhibit an increased binding affinity for the protein and a higher potential as DHODH inhibitors. Collectively, the elevated docking scores of these two new compounds can be attributed to their interactions with a greater number of key residues compared to the original compounds, which lacked such interactions. Moreover, the ring structure introduced by substitution facilitates the formation of more stable interactions. In addition, the DUH (crystal inhibitor) also performed docking and was shown in S3 Fig. It can be seen that two hydrogen bond interactions were formed, and the final docking score was −15.6344.

### 3.6 Molecular dynamics analysis

In Molecular Dynamics (MD) simulations, the Root Mean Square Deviation (RMSD) metric is critical for quantifying conformational similarities or divergences between molecular systems, serving as a key indicator to evaluate conformational dynamics over simulation trajectories. As illustrated in Fig 5A, during a 200 ns MD simulation, the conformational

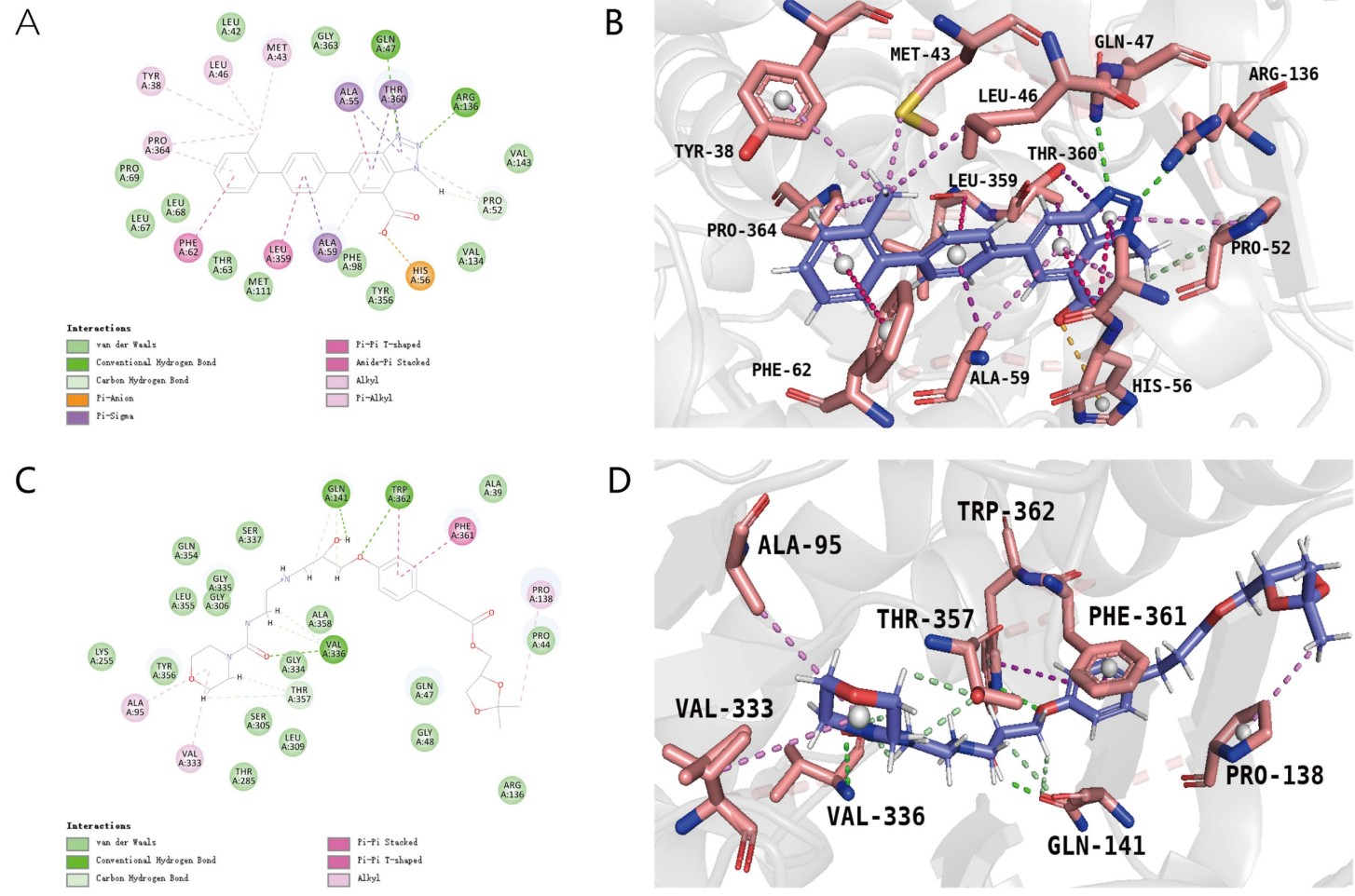

**Fig 3. 2D interaction diagram and 3D interaction diagram of the original compound 1281 and the positive control AG636. (A)** 2D interaction plot of the original compound 1281. Hydrogen bonds are shown as green dashed lines, Hydrophobicity is in red lines. **(B)** 3D interaction plot of the original compound 1281. Hydrogen bonds are shown as purple dashed lines. **(C)** 2D interaction plot of the positive control AG636. Hydrogen bonds are shown as green dashed lines, and Hydrophobicity is in red lines. **(D)** 3D interaction plot of the positive control AG636. Hydrogen bonds are shown as purple dashed lines.

fluctuations of both molecules remained relatively stable, primarily oscillating within a range of 0.15–0.3 nm. Notably, their RMSD values consistently stayed below 0.25 nm until the 180 ns time point, displaying a highly consistent trend that underscores more stable structural interactions with the target molecule. However, divergence emerged in the final 20 ns: the RMSD of compound 65:[(4S)-2,2-dimethyl-1,3-dioxolan-4-yl]methyl 3-(4-{[(2S)-2-hydroxypropyl]oxy}phenyl) increased and plateaued at 0.23 nm, whereas that of compound 66: [(4S)-2,2-dimethyl-1,3-dioxolan-4-yl]methyl 4-(4-{[(2S)-2-hydroxypropyl]oxy}phenyl) decreased and stabilized at 0.21 nm. In addition, the final RMSD value of the DUH remained stable at around 0.4nm In the second MD simulation, the two systems initiated stabilization by 150 ns and ultimately converged to a similar stabilized state at approximately 0.3 nm (S4 Fig). For the third simulation, however, the RMSD trajectories exhibited minor deviations: during the final 30 ns, the compound 66: [(4S)-2,2-dimethyl-1,3-dioxolan-4-yl] methyl 4-(4-{[(2S)-2-hydroxypropyl]oxy}phenyl) system remained stable at 0.25 nm, whereas the compound 65:[(4S)-2,2-dimethyl-1,3-dioxolan-4-yl]methyl 3-(4-{[(2S)-2-hydroxypropyl]oxy}phenyl) system maintained a stable RMSD of 0.2 nm (S5 Fig). Collectively, both complexes attained conformational stability by the conclusion of the simulation.

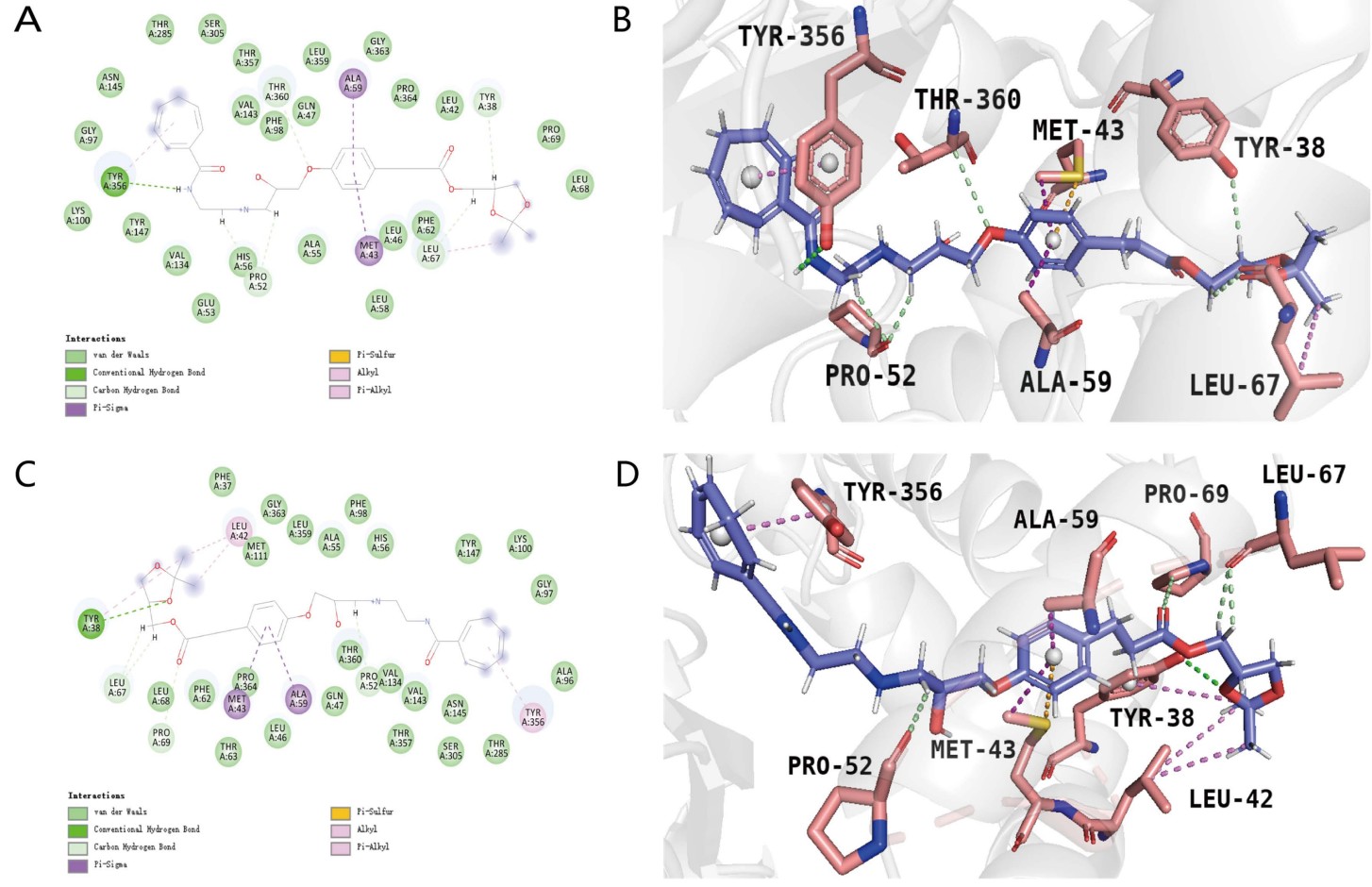

**Fig 4. 2D interaction diagram and 3D interaction diagram of compound 1281_65 and compound 1281_66. (A)** 2D interaction diagram of compound 1281_65. Hydrogen bonds are shown as green dashed lines, Hydrophobicity is in red lines. **(B)** 3D interaction diagram of compound 1281_65. Hydrogen bonds are shown as purple dashed lines. **(C)** 2D interaction diagram of compound 1281_66. Hydrogen bonds are shown as green dashed lines, Hydrophobicity is in red lines. **(D)** 3D interaction diagram of compound 1281_66. Hydrogen bonds are shown as purple dashed lines.

The Root Mean Square Fluctuation (RMSF) metric quantifies the positional variability of individual atoms or residues within a protein or molecular system during simulation, serving as a key indicator for identifying flexible regions—critical for elucidating protein functionality and stability. As depicted in Fig 5B, the RMSF profiles of both systems oscillated within a narrow range of 0.05–0.41 nm, validating the robustness of the simulation. Notably, a distinct divergence emerged at residue 300, proximal to the binding pocket: compound 65:[(4S)-2,2-dimethyl-1,3-dioxolan-4-yl]methyl 3-(4-{[(2S)-2-hydroxypropyl]oxy}phenyl) exhibited an RMSF peak of 0.4 nm at this site, whereas compound 66: [(4S)-2,2-dimethyl-1,3-dioxolan-4-yl]methyl 4-(4-{[(2S)-2-hydroxypropyl]oxy}phenyl) and DUH maintained a lower RMSF of 0.35 nm. In the RMSF analyses of the second and third simulations, the overall system exhibited RMSF fluctuations within a consistent range of 0.05–0.4 nm. Notably, the binding pocket region in the second simulation displayed a comparable RMSF trend to that of the first simulation. Specifically, system 66: [(4S)-2,2-dimethyl-1,3-dioxolan-4-yl]methyl 4-(4-{[(2S)-2-hydroxypropyl]oxy} phenyl) demonstrated a lower RMSF value (0.17 nm) at this site, indicating enhanced conformational stability relative to its counterpart (S4 Fig). Conversely, in the third simulation, the binding pocket residue of system 65:[(4S)-2,2-dimethyl-1,3-dioxolan-4-yl]methyl 3-(4-{[(2S)-2-hydroxypropyl]oxy}phenyl) exhibited an even lower RMSF value (0.16 nm), further

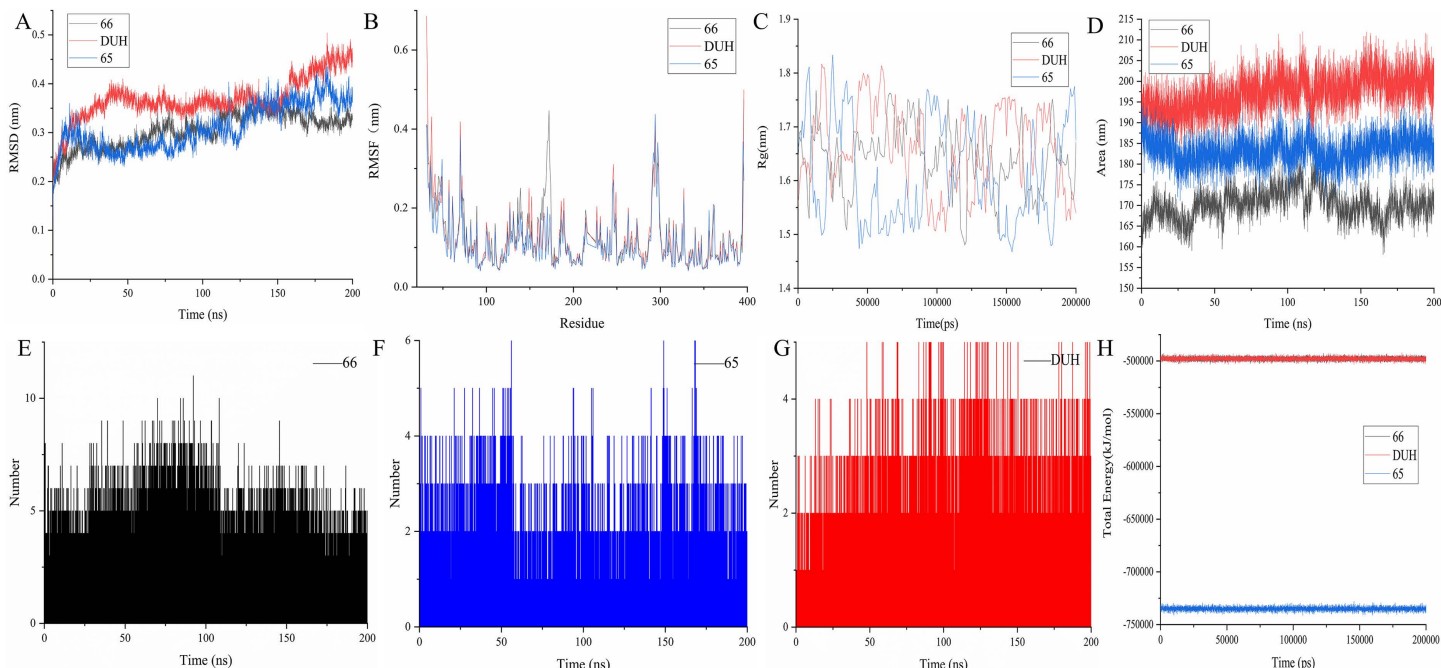

**Fig 5. Molecular dynamics analysis of compounds 65(blue) and 66(black) and DUH(red) with DHODH. (A)** The RMSD of compounds 65 and 66
、DUH with DHODH. Compound 65 is shown as blue lines, compound 66 is in the black line. DUH is in the red line. **(B)** The RMSF of compounds 65
and 66 and DUH with DHODH. Compound 65 is shown as blue lines, compound 66 is in the black line. DUH is in the red line. **(C)** The Rg of compounds
65 and 66 and DUH with DHODH. Compound 65 is shown as blue lines, compound 66 is in the black line. DUH is in the red line. **(D)** The Area of
compounds 65 and 66 and DUH with DHODH. Compound 65 is shown as blue lines, compound 66 is in the black line. DUH is in the red line. **(E)** The
Hydrogen bond number of compound 65 with DHODH. **(F)** The Hydrogen bond number of compound 66 with DHODH. **(G)** The Hydrogen bond number
of compound DUH with DHODH. **(H)** The total energy of compounds 65 and 66 and DUH with DHODH. Compound 65 is shown as blue lines, compound
66 is in the black line. DUH is in the red line.

highlighting its structural rigidity under these conditions (S5 Fig). Ultimately, both systems stabilized within a consistent
RMSF range, underscoring their conformational equilibration during the simulation.

The Radius of Gyration (Rg) is a significant parameter that characterizes the compactness of protein structures. A
smaller Rg indicates a more compact and stable protein conformation. Throughout the 100 ns simulation, the Rg val-
ues for both systems 65:[(4S)-2,2-dimethyl-1,3-dioxolan-4-yl]methyl 3-(4-{[(2S)-2-hydroxypropyl]oxy}phenyl) and 66:
[(4S)-2,2-dimethyl-1,3-dioxolan-4-yl]methyl 4-(4-{[(2S)-2-hydroxypropyl]oxy}phenyl) and DUH were consistently below
2.0 nm, as illustrated in Fig 5C, indicating that both systems maintained a consistent density. In the Rg analyses of the
second and third simulations, both systems consistently maintained their radius of gyration within a narrow range of
1.96–2.0 nm (S4 Fig). Notably, the third simulation revealed a trend consistent with the RMSF and RMSD data: specifi-
cally, the Rg of System 66: [(4S)-2,2-dimethyl-1,3-dioxolan-4-yl]methyl 4-(4-{[(2S)-2-hydroxypropyl]oxy}phenyl) exceeded
that of System 65:[(4S)-2,2-dimethyl-1,3-dioxolan-4-yl]methyl 3-(4-{[(2S)-2-hydroxypropyl]oxy}phenyl) and DUH, mirroring
the comparative patterns observed in the earlier metrics (S5 Fig).

The term "Area" in this context typically refers to the Solvent Accessible Surface Area (SASA), a key biophysical param-
eter for characterizing the surface area of proteins or molecules that interacts with a solvent (e.g., water). As illustrated in
Fig 5D, during the 200 ns MD simulation, the SASA values of both systems exhibited minimal variability, remaining stably
within the range of 160–190 nm². Similarly, in the second and third simulations, the Area value remained within the range
of 180-200nm². However, the range of crystal inhibitors exceeds 200 nm² (S4 and S5 Figs).

Hydrogen bonds are critical mediators of interactions between proteins and solvent molecules, with their quantity and spatial distribution serving as key determinants of molecular stability, structural integrity, and functional properties. As depicted in Fig 5E–5G, Compound 65:[(4S)-2,2-dimethyl-1,3-dioxolan-4-yl]methyl 3-(4-{[(2S)-2-hydroxypropyl]oxy}phenyl) maintained more than two hydrogen bonds throughout the 200 ns MD simulation, whereas Compound 66: [(4S)-2,2-dimethyl-1,3-dioxolan-4-yl]methyl 4-(4-{[(2S)-2-hydroxypropyl]oxy}phenyl) exhibited a higher number of such interactions. This discrepancy in hydrogen bond formation suggests a greater degree of conformational stability in the Compound 66: [(4S)-2,2-dimethyl-1,3-dioxolan-4-yl]methyl 4-(4-{[(2S)-2-hydroxypropyl]oxy}phenyl) system. In the second and third MD simulations, the hydrogen bond count of System 66: [(4S)-2,2-dimethyl-1,3-dioxolan-4-yl]methyl 4-(4-{[(2S)-2-hydroxypropyl]oxy}phenyl) remained consistently higher and more persistent compared to that of System 65:[(4S)-2, 2-dimethyl-1,3-dioxolan-4-yl]methyl 3-(4-{[(2S)-2-hydroxypropyl]oxy}phenyl), suggesting that the System 66: [(4S)-2, 2-dimethyl-1,3-dioxolan-4-yl]methyl 4-(4-{[(2S)-2-hydroxypropyl]oxy}phenyl) complex exhibited superior performance to System 65:[(4S)-2,2-dimethyl-1,3-dioxolan-4-yl]methyl 3-(4-{[(2S)-2-hydroxypropyl]oxy}phenyl) in both hydrogen bond formation capacity and conformational stability (S4 and S5 Figs).

The total energy, defined as the sum of potential and kinetic energies of all atoms in the system, serves as a fundamental descriptor of thermodynamic state and stability. As depicted in Fig 5H, both compounds 65:[(4S)-2,2-dimethyl-1,3-dioxolan-4-yl]methyl 3-(4-{[(2S)-2-hydroxypropyl]oxy}phenyl) and 66: [(4S)-2,2-dimethyl-1,3-dioxolan-4-yl]methyl 4-(4-{[(2S)-2-hydroxypropyl]oxy}phenyl) exhibited a plateau at approximately $-7.400 \times 10^6$ kJ mol$^{-1}$ throughout the 200 ns MD trajectory, signifying that the systems had achieved equilibrium. In contrast, the subsequent replicate simulations revealed marked divergences: system 65:[(4S)-2,2-dimethyl-1,3-dioxolan-4-yl]methyl 3-(4-{[(2S)-2-hydroxypropyl]oxy}phenyl) converged to $-7.400 \times 10^5$ kJ mol$^{-1}$, whereas system 66: [(4S)-2,2-dimethyl-1,3-dioxolan-4-yl]methyl 4-(4-{[(2S)-2-hydroxypropyl]oxy}phenyl) stabilized at a markedly lower value of $-7.410 \times 10^5$ kJ mol$^{-1}$. These results indicate that system 66: [(4S)-2,2-dimethyl-1,3-dioxolan-4-yl]methyl 4-(4-{[(2S)-2-hydroxypropyl]oxy}phenyl) possesses superior thermodynamic stability relative to system 65:[(4S)-2,2-dimethyl-1,3-dioxolan-4-yl]methyl 3-(4-{[(2S)-2-hydroxypropyl]oxy}phenyl), DUH exhibited a plateau at approximately $-5.100 \times 10^6$ kJ mol$^{-1}$ throughout the 200 ns MD trajectory (S4 and S5 Figs).

### 3.7 MM-PBSA analysis

MM-PBSA (Molecular Mechanics-Poisson-Boltzmann Surface Area) is a computational approach for estimating the binding free energy of molecular complexes. This method leverages molecular dynamics (MD) simulation trajectories to calculate the energy of molecular systems. By modeling the solvent as a continuum and integrating molecular mechanics, Poisson-Boltzmann equations, and surface area calculations, MM-PBSA provides an estimation of the binding free energy. Compounds 65:[(4S)-2,2-dimethyl-1,3-dioxolan-4-yl]methyl 3-(4-{[(2S)-2-hydroxypropyl]oxy}phenyl) and 66: [(4S)-2,2-dimethyl-1,3-dioxolan-4-yl]methyl 4-(4-{[(2S)-2-hydroxypropyl]oxy}phenyl) were chosen for MM-PBSA analysis based on their demonstrated structural stability, flexibility, and favorable interaction simulation outcomes. Throughout a 200 ns molecular dynamics (MD) simulation, the binding free energy components of the two protein-ligand complexes were systematically calculated at 1 ns intervals. As summarized in Table 6, the MM-PBSA-derived binding free energies for complexes of compound 65:[(4S)-2,2-dimethyl-1,3-dioxolan-4-yl]methyl 3-(4-{[(2S)-2-hydroxypropyl]oxy}phenyl) and 66: [(4S)-2,2-dimethyl-1,3-dioxolan-4-yl]methyl 4-(4-{[(2S)-2-hydroxypropyl]oxy}phenyl) and DUH with DHODH were determined to be −145 kJ/mol and −150 kJ/mol、-135kj/mol, respectively (both significantly below thermodynamic thresholds for stable binding), with van der Waals interactions emerging as the dominant energetic contribution. These strongly negative binding free energy values, validated by MM-PBSA analysis, indicate robust and stable ligand-protein interactions. To gain mechanistic insights into the molecular determinants of these interactions, we decomposed the MM-PBSA binding free energy contributions to quantify the energetic role of individual protein residues, thereby identifying key binding residues—those primarily responsible for driving the overall binding affinity. As illustrated in Fig 6A, the key residues mediating the interaction between compound 65:[(4S)-2,2-dimethyl-1,3-dioxolan-4-yl]methyl 3-(4-{[(2S)-2-hydroxypropyl]

**Table 6. The binding energy of binding for compounds 65 and 66、 DUH with DHODH.**

| Molecules | Van der Waal energy kJ/mol | Electrostatic energy kJ/mol | Polar solvation energy kJ/mol | SASA energy kJ/mol | Binding energy kJ/mol |
|---|---|---|---|---|---|
| 65 1st | −300.787 +/-12.988 | −245.485 +/- 22.132 | 126.479 +/-26.866 | −29.476 +/-1.112 | −203.757 +/-4.495 |
| 65 2nd | −266.808 +/- 87.825 | −189.069 +/-38.653 | 237.214 +/-73.683 | −26.781 +/-8.886 | −245.444 +/61.681 |
| 65 3rd | −229.665 +/-76.487 | −246.798 +/-66.079 | 109.128 +/-54.509 | −25.795 +/-8.991 | −146.331 +/-24.018 |
| 66 1st | −105.447 +/-134.538 | −237.169 +/-101.469 | 45.127 +/-71.257 | −10.739 +/-14.089 | −171.058 +/-69.925 |
| 66 2nd | −291.192 +/-13.225 | −301.326 +/-33.537 | 115.755 +/-40.090 | −30.053 +/-1.087 | −205.46 +/-17.211 |
| 66 3rd | −283.171 +/-13.537 | −245.860 +/-21.403 | 160.722 +/-31.713 | −28.098 +/-1.088 | −150.546 +/-6.855 |
| DUH 1st | −259.19 +/-18.143 | −106.839 +/-35.526 | 240.514 +/-73.113 | −9.639 +/-12.089 | −135.154 +/-7.355 |
| DUH 2nd | −276.249 +/-20.541 | −83.008 +/-22.162 | 225.434 +/-63.653 | −6.731 +/-11.659 | −140.554 +/-5.155 |
| DUH 3rd | −265.319 +/-24.423 | −108.336 +/-10.489 | 234.264 +/-58.563 | −7.459 +/-13.729 | −137.850 +/-6.225 |

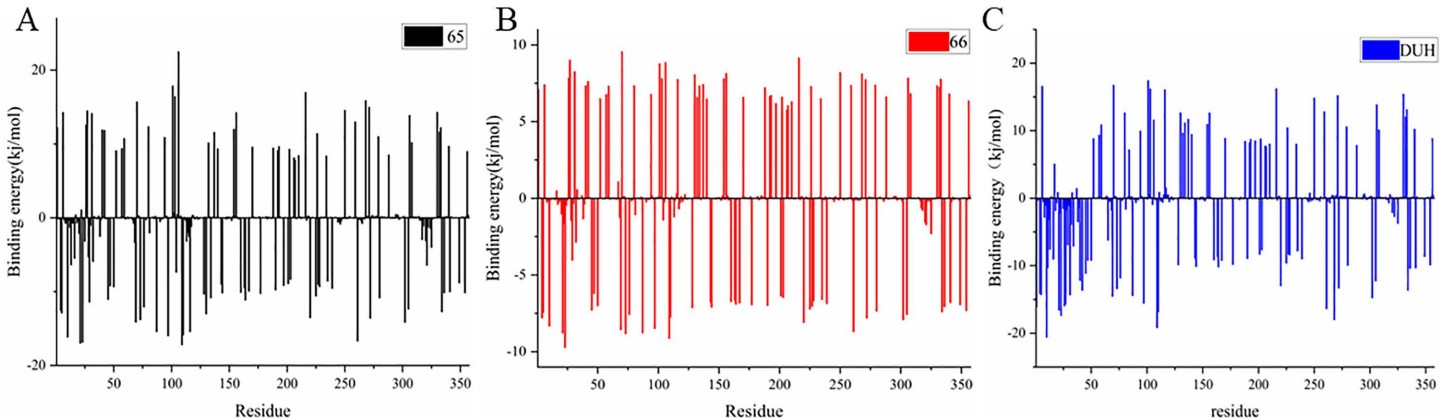

**Fig 6. Residue decomposition diagram of binding energy. (A)** Compound 65 with DHODH (black). **(B)** Compound 66 with DHODH (red). **(C)** Compound DUH with DHODH (blue).

oxy}phenyl) and DHODH were identified as Leu-67, Tyr-38, Met-43, Ala-59, and Tyr-356. Notably, these residues overlap with those implicated in hydrogen bonding networks observed in molecular docking simulations. For compound 66: [(4S)-2,2-dimethyl-1,3-dioxolan-4-yl]methyl 4-(4-{[(2S)-2-hydroxypropyl]oxy}phenyl), the DHODH-interacting key residues were expanded to include Tyr-38, Ala-59, Leu-42, Leu-67, Pro-69, Met-43, and Tyr-356 (Fig 6B). For compound DUH, the DHODH-interacting key residues were expanded to include Tyr-38, Ala-59, Tyr-38, Met-43, Leu-42, Leu-67, Pro-69, and Tyr-356 (Fig 6C).Collectively, these MM-PBSA results not only corroborate the molecular docking predictions but also provide a quantitative assessment of the energetic contributions underpinning the binding of compounds 65:[(4S)-2,2-dimethyl-1,3-dioxolan-4-yl]methyl 3-(4-{[(2S)-2-hydroxypropyl]oxy}phenyl) and 66: [(4S)-2,2-dimethyl-1,3-dioxolan-4-yl]methyl 4-(4-{[(2S)-2-hydroxypropyl]oxy}phenyl) and DUH to DHODH (S6 and S7 Figs). This integrated analysis strengthens our understanding of the structural and energetic basis of ligand-receptor recognition for these small-molecule inhibitors.

## 3.8 Principal component analysis

PCA (Principal Component Analysis) is a statistical method used to discover patterns in data by transforming a set of potentially correlated variables into a set of linearly uncorrelated variables called principal components through orthogonal transformations. The purpose of PCA is to reduce the dimensionality of the data while retaining as much information as

possible from the original data. By focusing on these principal components, the dynamics of the system are easier to visualize and understand. In evaluating MD simulations, PCA is widely used in data preparation, covariance matrix calculation, eigenvalue decomposition, principal component projection, analysis, interpretation, protein dynamics analysis, protein folding studies, and ligand binding studies due to its advantages of dimensionality reduction, collective motion recognition, and dynamic display. Thus PCA is an extremely adaptable and powerful method for evaluating molecular dynamics simulations. During MD simulations, two-dimensional projections of trajectories onto PC1 and PC2 provide a clear and concise picture of the structural dynamics of the complexes. The dispersion of the plots shows a positive correlation with conformational diversity. As the data points on the PC projections are more widely distributed, it indicates that a wider range of conformational states of the system emerges during the simulation. In Fig 7, the eigenvalues of the proteins in the 65 system exhibit an exponential correlation with the eigenvectors corresponding to the first six motion modes. Scatter plots illustrate the

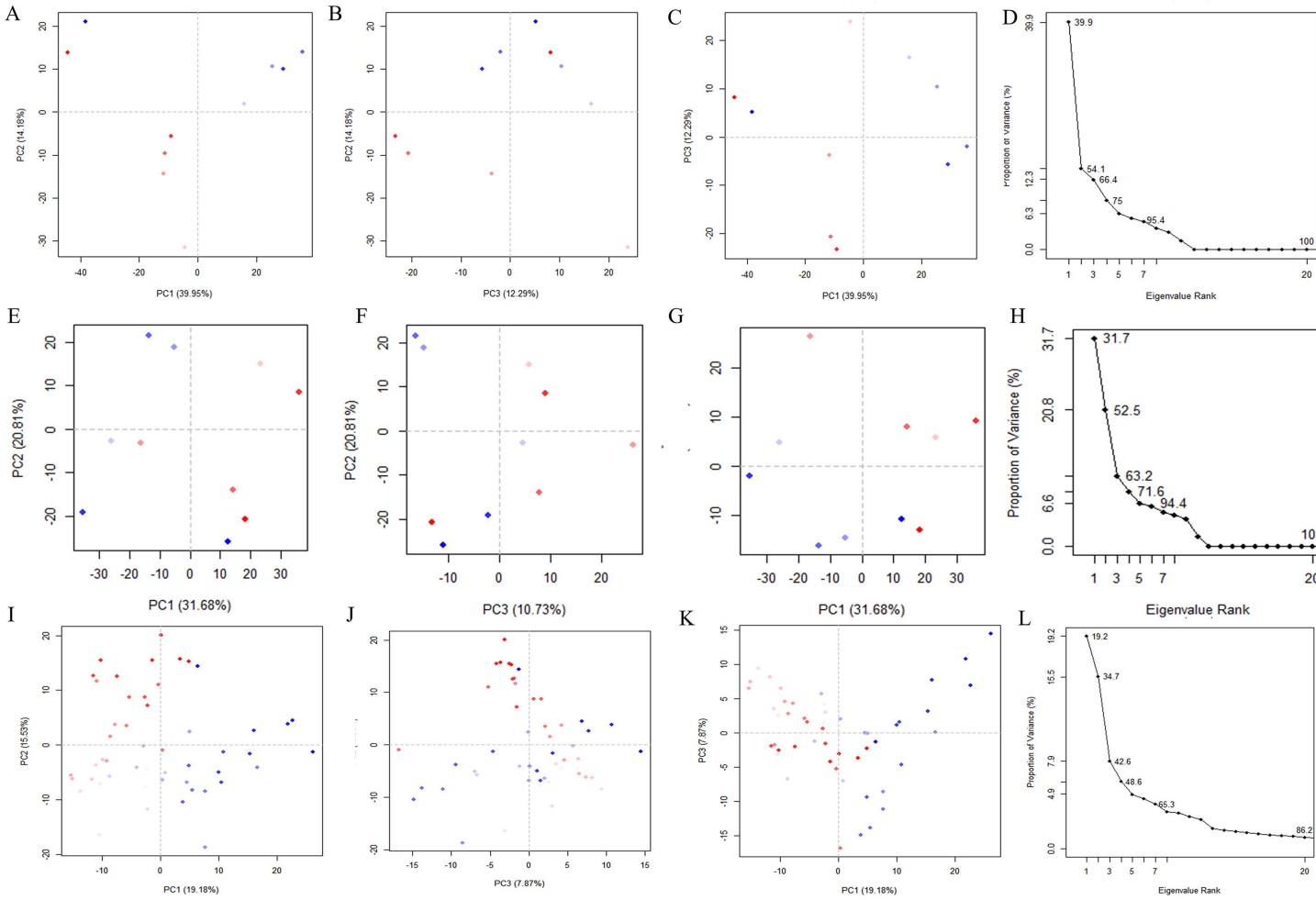

**Fig 7. Principal component analysis plot for the complex 65 and 66 and DUH. (A-L)** The Fig shows the scatter plot among the first three principal components (PC1, PC2, and PC3) of the 65 system, with the blue region exhibiting the most pronounced motion, the white region exhibiting intermediate motion, and the red region exhibiting lesser motion. **(E-H)** The Fig shows the scatter plot among the first three principal components (PC1, PC2, and PC3) of the 66 system, with the blue region exhibiting the most pronounced motion, the white region exhibiting intermediate motion, and the red region exhibiting lesser motion. **(I-L)**The Fig shows the scatter plot among the first three principal components (PC1, PC2, and PC3) of the DUH system, with the blue region exhibiting the most pronounced motion, the white region exhibiting intermediate motion, and the red region exhibiting lesser motion.

relationships among the first three principal components (PC1, PC2, and PC3), which collectively account for 66.4% of the total motion variance in the system. Similarly, the first three principal components of the 66: [(4S)-2,2-dimethyl-1,3-dioxolan-4-yl]methyl 4-(4-{[(2S)-2-hydroxypropyl]oxy}phenyl) system explain 63.2% of the total motion variance. These findings suggest that the intramolecular dynamics of both systems resemble critical conformational events such as protein folding or ligand-induced conformational transitions. Within the 65:[(4S)-2,2-dimethyl-1,3-dioxolan-4-yl]methyl 3-(4-{[(2S)-2-hydroxypropyl]oxy}phenyl) system, the PC1 cluster exhibits the highest variability (39.95%), followed by PC2 (14.18%) and PC3 (12.29%). For the 66: [(4S)-2,2-dimethyl-1,3-dioxolan-4-yl]methyl 4-(4-{[(2S)-2-hydroxypropyl]oxy}phenyl) system, the variability of the PC1, PC2, and PC3 clusters is 31.68%, 20.81%, and 10.73%, respectively. Notably, PC1 consistently shows the highest variability across both systems, while PC3 displays the smallest variability—a pattern consistent with the third principal component often being associated with the most stable protein-ligand binding interactions. This trend is further validated in the second and third simulation trajectories (S7 and S8 Figs). To further evaluate protein-ligand interactions, we analyzed the covariance matrix (Fig 8), where dark blue regions denote synergistic motions and light yellow regions indicate relatively independent motions. The 66: [(4S)-2,2-dimethyl-1,3-dioxolan-4-yl]methyl 4-(4-{[(2S)-2-hydroxypropyl]oxy}phenyl) complex exhibits more pronounced synergistic effects in its motion patterns, aligning with previous observations such as lower RMSD values. Collectively, these results confirm that the first three principal components of both the 65:[(4S)-2,2-dimethyl-1,3-dioxolan-4-yl]methyl 3-(4-{[(2S)-2-hydroxypropyl]oxy}phenyl) and 66: [(4S)-2,2-dimethyl-1,3-dioxolan-4-yl]methyl 4-(4-{[(2S)-2-hydroxypropyl]oxy}phenyl) complexes account for less than 45% of the total motion variance, indicating stable conformational behavior throughout the simulation period. The first three principal components of the DUH system explain 57.2%、52.2% and 42.6% of the total motion variance (S8 and S9 Figs).

### 3.9 Free energy landscape

During molecular dynamics (MD) simulations, free energy landscape analysis provides a comprehensive visualization of energy distribution and conformational dynamics within biomolecular complexes. This analytical approach facilitates the characterization of diverse conformations and energy minima adopted by protein-ligand complexes during their dynamic functions, offering critical insights into the thermodynamic and kinetic properties governing their behavior.A three-dimensional free energy landscape is typically constructed using the first two principal components (PC1 and PC2) derived from principal component analysis (PCA), which effectively reduces the high-dimensional conformational data into a lower-dimensional space. This landscape is instrumental for identifying conformational transitions, steady-state regions,

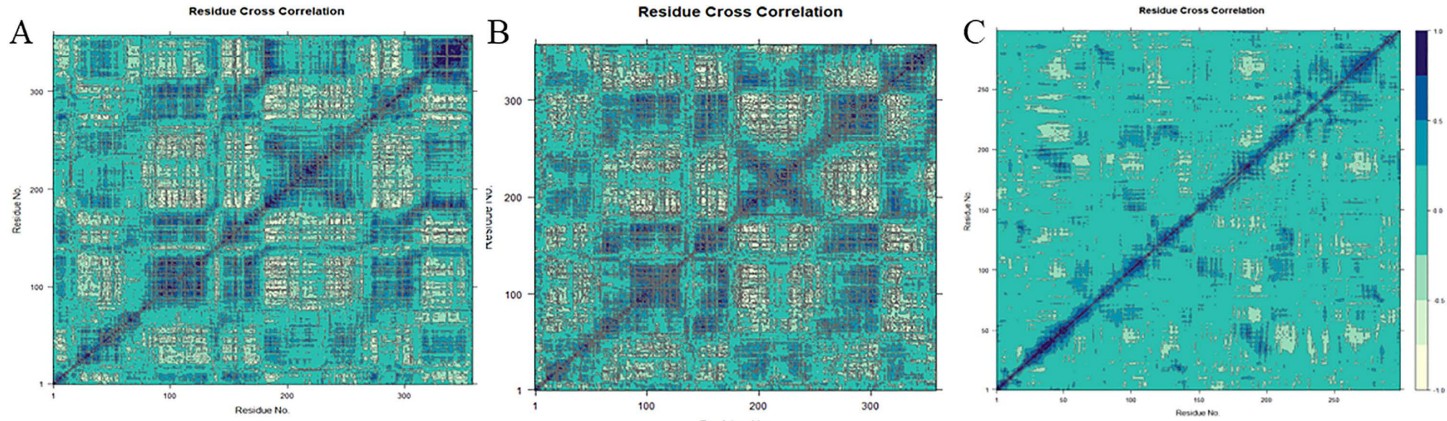

**Fig 8. Dynamic interrelationship diagram of the complex. (A)** Compound 65 with DHODH **(B)** Compound 66 with DHODH. **(C)** DUH with DHODH. Positive correlations between residues are shown in cyan and negative correlations are shown in light green.

and energy potential barriers: peaks correspond to high-energy transition states, while basins represent low-energy, stable conformations. Deciphering these features is central to understanding important interactions, conformational changes, and the overall stability of biomolecular systems, as PCA simplifies complex conformational data into an interpretable representation of the free energy environment. Such analysis is particularly valuable for investigating biological processes such as protein-ligand interactions and macromolecular conformational transitions. The 3D projection plots (as shown in Fig 9) visually depict dynamic conformational changes occurring during the simulation, with dark blue regions in the energy landscape highlighting areas of lowest energy—for example, the formation of low-energy structures resembling narrow funnels. These basin structures correspond to protein conformations that reach minimal energy levels at specific simulation timepoints, with darker blue regions denoting more stable local energy minima. Notably, the energy landscapes of complexes 65:[(4S)-2,2-dimethyl-1,3-dioxolan-4-yl]methyl 3-(4-{[(2S)-2-hydroxypropyl]oxy}phenyl) and 66: [(4S)-2,2-dimethyl-1,3-dioxolan-4-yl]methyl 4-(4-{[(2S)-2-hydroxypropyl]oxy}phenyl) and DUH each exhibit distinct basin structures, indicating well-defined, stable conformational states. Additionally, these landscapes can be categorized into "narrow basin" and "wide basin" types: molecules in wide basins generally display greater conformational stability compared to those in narrow basins. The presence of well-resolved basins in both systems suggests enhanced binding stability of complexes 65:[(4S)-2,2-dimethyl-1,3-dioxolan-4-yl]methyl 3-(4-{[(2S)-2-hydroxypropyl]oxy}phenyl) and 66: [(4S)-2,2-dimethyl-1,3-dioxolan-4-yl]methyl 4-(4-{[(2S)-2-hydroxypropyl]oxy}phenyl) and DUH with DHODH (S12 and S13 Figs). Integration of these kinetic and principal component analysis results further supports the conclusion that both complexes form stable bindings with DHODH, providing a robust foundation for understanding their molecular recognition mechanisms.

## 4. Discussion

Prior research has demonstrated that dihydroorotate dehydrogenase (DHODH), a crucial mitochondrial enzyme required for the de novo synthesis of pyrimidines, is frequently overexpressed in numerous malignancies, such as acute myeloid leukemia (AML), breast cancer, colorectal cancer (CRC), and melanoma. Further investigations have revealed a significant correlation between elevated DHODH expression in various cancer cells and ferroptosis, a form of regulated cell death. Consequently, the inhibition of DHODH presents a promising therapeutic strategy for cancer treatment. Computer-aided drug design (CADD) significantly expedites the process and conserves resources, enabling researchers to identify lead compounds rapidly. Rescreening the FDA-approved drug pool presents a viable strategy to resolve this dilemma.

This study performed a virtual screening of the FDA database based on known inhibitor re-purposing techniques. The known DHODH inhibitor AG636 was used as a control ligand for comparative analysis, and possible candidates for DHODH inhibition were identified. The results showed that compounds 4410,1792, 6043, 1281, 1282 and 1106 had higher binding affinity to the target protein than the control. The binding scores of these compounds were all greater than -20 kcal/mol, which was higher than that of the control compound AG636 at -17.6145 kcal/mol. The best results were obtained for the docking of compounds 65:[(4S)-2,2-dimethyl-1,3-dioxolan-4-yl]methyl 3-(4-{[(2S)-2-hydroxypropyl]oxy} phenyl) and 66: [(4S)-2,2-dimethyl-1,3-dioxolan-4-yl]methyl 4-(4-{[(2S)-2-hydroxypropyl]oxy}phenyl) after fragment substitution, in which compound 65:[(4S)-2,2-dimethyl-1,3-dioxolan-4-yl]methyl 3-(4-{[(2S)-2-hydroxypropyl]oxy}phenyl) formed a hydrogen bond with Thr-360 and compound 66: [(4S)-2,2-dimethyl-1,3-dioxolan-4-yl]methyl 4-(4-{[(2S)-2-hydroxypropyl] oxy}phenyl) formed a hydrogen bond with Tyr-3, which were maintained even after the MD simulation of the complexes. RMSD plots showed that the final MD trajectory conformation was stable, with compound 65:[(4S)-2,2-dimethyl-1,3-dioxolan-4-yl]methyl 3-(4-{[(2S)-2-hydroxypropyl]oxy}phenyl) remaining at 0.25 nm and compound 66: [(4S)-2,2-dimethyl-1,3-dioxolan-4-yl]methyl 4-(4-{[(2S)-2-hydroxypropyl]oxy}phenyl) remaining below 0.23 nm. There was a change in the protein conformation of any of the complexes during the simulation. However, the conformational stability of 65:[(4S)-2,2-dimethyl-1,3-dioxolan-4-yl]methyl 3-(4-{[(2S)-2-hydroxypropyl]oxy}phenyl) and 66: [(4S)-2,2-dimethyl-1,3-dioxolan-4-yl]methyl 4-(4-{[(2S)-2-hydroxypropyl]oxy}phenyl) was higher in the later stages of the simulation. This also indicates that the docking sites of these compounds are accurate. compounds 65:[(4S)-2,2-dimethyl-1,3-dioxolan-4-yl]

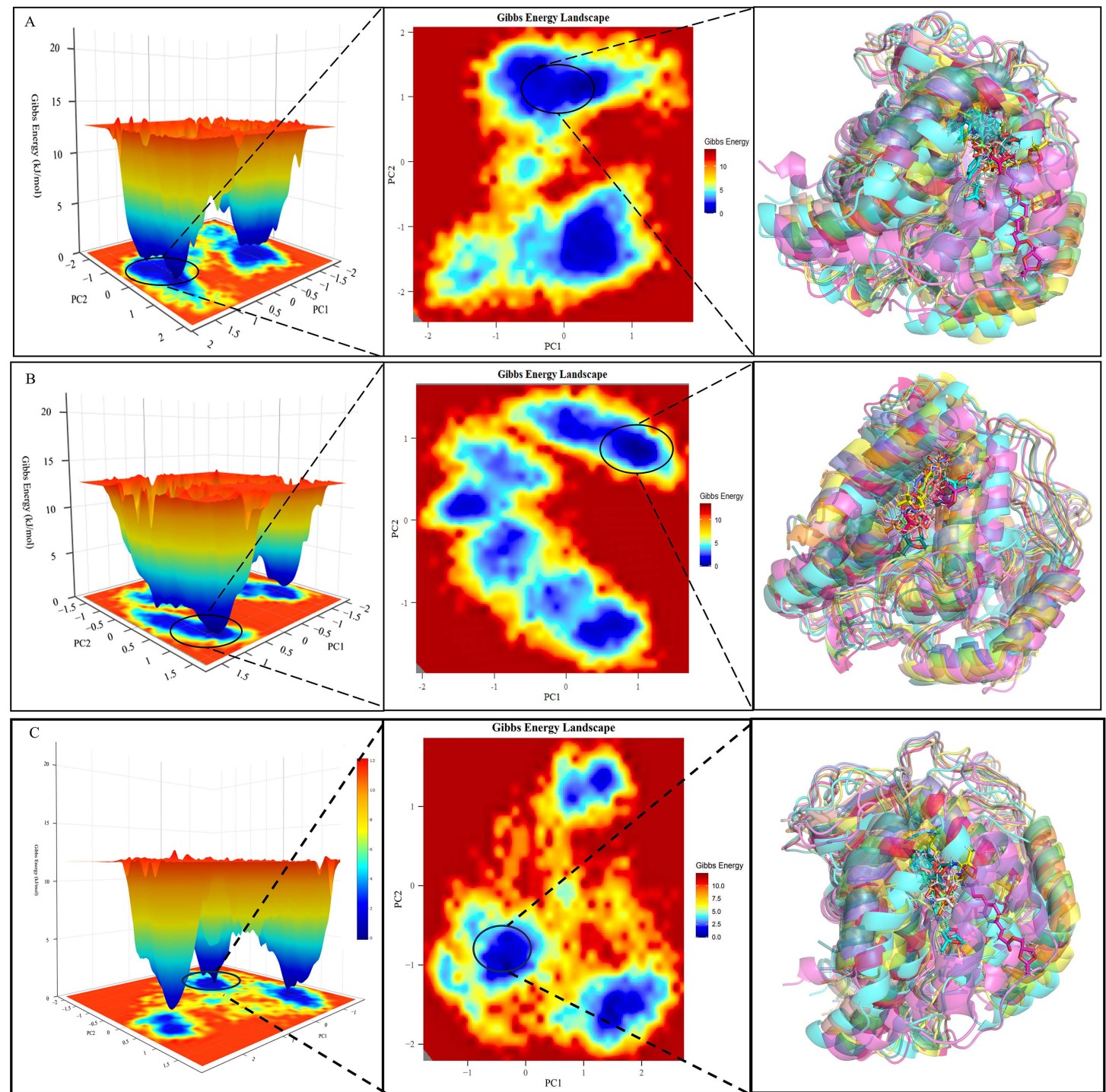

**Fig 9. PCA-based Free-energy topography and variation of the lowest energy conformation of the protein-ligand. (A)** Compound 65 with DHODH **(B)** Compound 66 with DHODH. **(C)** DUH with DHODH.

methyl 3-(4-{[(2S)-2-hydroxypropyl]oxy}phenyl) and 66: [(4S)-2,2-dimethyl-1,3-dioxolan-4-yl]methyl 4-(4-{[(2S)-2-hydroxypropyl]oxy}phenyl) can be considered as potential candidates within the DHODH binding site. As for the RMSF, the RMSF values of the key residues in the docking pocket are all less than 0.4 nm. Removing the terminal residues, which are supposed to be stable during the simulation process. The RMSF plots show the stable conformation of all the residues in the binding site. This indicates that the ligands present in the binding site do not show significant instability in the protein conformation.

Hydrogen bonding analysis during MD simulation shows that compounds 65:[(4S)-2,2-dimethyl-1,3-dioxolan-4-yl]methyl 3-(4-{[(2S)-2-hydroxypropyl]oxy}phenyl) and 66: [(4S)-2,2-dimethyl-1,3-dioxolan-4-yl]methyl 4-(4-{[(2S)-2-hydroxypropyl]oxy}phenyl) have significant binding to the protein during 200ns. The total MMPBSA binding free energies of -245.444kj/mol and -205.46kj/mol for compounds 65:[(4S)-2,2-dimethyl-1,3-dioxolan-4-yl]methyl 3-(4-{[(2S)-2-hydroxypropyl]oxy}phenyl) and 66: [(4S)-2,2-dimethyl-1,3-dioxolan-4-yl]methyl 4-(4-{[(2S)-2-hydroxypropyl]oxy}phenyl), respectively, indicate a strong binding affinity with the protein. In comparison, compound 66: [(4S)-2,2-dimethyl-1,3-dioxolan-4-yl]methyl 4-(4-{[(2S)-2-hydroxypropyl]oxy}phenyl) exhibited the best binding energy value.

PCA and FEL were used to analyze the stability of components and conformations during specific simulations. The dispersion in the PCA plot reflects the conformational variability of the structure. Among them, the PC values of complexes 65:[(4S)-2,2-dimethyl-1,3-dioxolan-4-yl]methyl 3-(4-{[(2S)-2-hydroxypropyl]oxy}phenyl) and 66: [(4S)-2,2-dimethyl-1,3-dioxolan-4-yl]methyl 4-(4-{[(2S)-2-hydroxypropyl]oxy}phenyl) were 66.4 and 63.2, respectively, which indicated that neither of them underwent much spatial variation during the simulation process. In the free energy landscape map and the superimposed conformational map of the lowest energy, it can be clearly judged that the system of compound 66: [(4S)-2,2-dimethyl-1,3-dioxolan-4-yl]methyl 4-(4-{[(2S)-2-hydroxypropyl]oxy}phenyl) shows a more stable state than compound 65:[(4S)-2,2-dimethyl-1,3-dioxolan-4-yl]methyl 3-(4-{[(2S)-2-hydroxypropyl]oxy}phenyl) in this simulation process. In summary, in this study, three-dimensional free energy landscape analysis by PCA was used to investigate the dynamic conformational changes and energy distribution of ligand-bound complexes and both were compared. The visual representation of these landscapes reveals the emergence of narrow funnel-like structures and the presence of deep purple basins, suggesting a localized state of minimal energy and high stability. That is, compound 66: [(4S)-2,2-dimethyl-1,3-dioxolan-4-yl]methyl 4-(4-{[(2S)-2-hydroxypropyl]oxy}phenyl) is more suitable as the best inhibitor of DHODH in the present study.

For the original crystal inhibitors of the structure, in this study, we incorporated the entire simulation process and conducted comparisons from all aspects, that is, DUH was used as a strict positive control compound. Finally, it can be seen that the compounds selected from the docking scores and models show a tighter binding than DUH, and this also corresponds to the subsequent kinetic results and the numerical reflection of MMPBSA. Therefore, it proves from another aspect that the compounds selected in this study, after optimizing their structure, can be considered as lead compounds.

Therefore, this study uses molecular dynamics (MD) simulations and subsequent post-simulation analyses to examine the behavior of ligands in more sensitive environments. Moreover, this is a novel approach. These results confirm the originality of the present investigation. In computer-aided drug development (CADD), the occurrence of false positives is inherently inevitable. Although advanced computational methods such as molecular docking and virtual screening enable high-throughput compound library analysis (ligand-based virtual screening reduces experimental screening costs), limitations in computational models and the inherent complexity of target-ligand interactions necessitate experimental validation of candidate compounds' biological activity. For instance, while PAINS rules can identify certain false positives, their coverage of complex mechanisms remains limited and may incorrectly flag approved drugs. Therefore, an integrated "dry-wet" workflow combining computational predictions with experimental validation is critical for enhancing the efficiency of drug discovery.

## 5. Conclusions

As a key target of ferroptosis, DHODH could play an anti-tumor role. In this study, six small molecules were obtained from 6140 FDA drugs by combining virtual screening and pharmacophore models. These molecules were further optimized to

produce 86 new compounds. Precise docking and ADMET analysis confirmed the good pharmacological properties of molecules 65:[(4S)-2,2-dimethyl-1,3-dioxolan-4-yl]methyl 3-(4-{[(2S)-2-hydroxypropyl]oxy}phenyl) and 66: [(4S)-2,2-dimethyl-1,3-dioxolan-4-yl]methyl 4-(4-{[(2S)-2-hydroxypropyl]oxy}phenyl). Finally, molecular dynamics simulation showed that compound 66: [(4S)-2,2-dimethyl-1,3-dioxolan-4-yl]methyl 4-(4-{[(2S)-2-hydroxypropyl]oxy}phenyl) could be used as a novel DHODH inhibitor, which provided a new therapeutic option. This study provides theoretical guidance for the design and synthesis of subsequent inhibitors.

## Supporting information

**S1 Table. Small molecules that form pharmacophore models.**
(DOCX)

**S2 Table. Active molecule.**
(DOCX)

**S1 Fig. Spatial distribution diagrams.** (A, E) are the spatial distribution diagrams of the main components (PC 1-PC 6) of the compound: (B). Two-dimensional spatial distribution of PC1 and PC2, (C). Two-dimensional spatial distribution of PC1 and PC3, (D). Two-dimensional spatial distribution of PC2 and PC3, (F). Two-dimensional spatial distribution of PC4 and PC5, (G). Two-dimensional spatial distribution of PC4 and PC6, (H). Two-dimensional spatial distribution of PC5 and PC6.
(TIF)

**S2 Fig. Test ROC curves of the model.** (A): Ten-fold ROC curve of the Bayesian model (B): Multi-set validation ROC curves of the NB model.
(TIF)

**S3 Fig. 2D interaction diagram and 3D interaction diagram of compound DUH.** (A) 3D interaction diagram of com-poundDUH. Hydrogen bonds are shown as green dashed lines, Hydrophobicity is in red lines. (B) 2D interaction diagram of compound DUH. Hydrogen bonds are shown as purple dashed lines.
(TIF)

**S4 Fig. The second time Molecular dynamics analysis of compounds 65(blue) and 66 (black) DUH(red) with DHODH.** (A) The RMSD of compounds 65 and 66、DUH with DHODH. Compound 65 is shown as blue lines, compound 66 is in the black line. DUH is in the red line. (B) The RMSF of compounds 65 and 66 and DUH with DHODH. Compound 65 is shown as blue lines, compound 66 is in the black line. DUH is in the red line. (C) The Rg of compounds 65 and 66 and DUH with DHODH. Compound 65 is shown as blue lines, compound 66 is in the black line. DUH is in the red line. (D) The Area of compounds 65 and 66 and DUH with DHODH. Compound 65 is shown as blue lines, compound 66 is in the black line. DUH is in the red line. (E) The Hydrogen bond number of compound 65 with DHODH. (F) The Hydrogen bond number of compound 66 with DHODH. (G) The Hydrogen bond number of compound DUH with DHODH. (H) The total energy of compounds 65 and 66 and DUH with DHODH. Compound 65 is shown as blue lines, compound 66 is in the black line. DUH is in the red line.
(TIF)

**S5 Fig. The third time molecular dynamics analysis of compounds 65(blue) and 66 (black) DUH(red) with DHODH.** (A) The RMSD of compounds 65 and 66、DUH with DHODH. Compound 65 is shown as blue lines, compound 66 is in the black line. DUH is in the red line. (B) The RMSF of compounds 65 and 66 and DUH with DHODH. Compound 65 is shown as blue lines, compound 66 is in the black line. DUH is in the red line. (C) The Rg of compounds 65 and 66 and DUH with DHODH. Compound 65 is shown as blue lines, compound 66 is in the black line. DUH is in the red line. (D) The

Area of compounds 65 and 66 and DUH with DHODH. Compound 65 is shown as blue lines, compound 66 is in the black line. DUH is in the red line. (E) The Hydrogen bond number of compound 65 with DHODH. (F) The Hydrogen bond number of compound 66 with DHODH. (G) The Hydrogen bond number of compound DUH with DHODH. (H) The total energy of compounds 65 and 66 and DUH with DHODH. Compound 65 is shown as blue lines, compound 66 is in the black line. DUH is in the red line.
(TIF)

**S6 Fig. The second time Residue decomposition diagram of binding energy.** (A) Compound 65 with DHODH (black). (B) Compound 66 with DHODH (red). (C) Compound DUH with DHODH (blue).
(TIF)

**S7 Fig. The third time Residue decomposition diagram of binding energy.** (A) Compound 65 with DHODH (black). (B) Compound 66 with DHODH (red). (C) Compound DUH with DHODH (blue).
(TIF)

**S8 Fig. The second time Principal component analysis plot for the complex 65 and 66 and DUH.** (A-L) The Fig shows the scatter plot among the first three principal components (PC1, PC2, and PC3) of the 65 system, with the blue region exhibiting the most pronounced motion, the white region exhibiting intermediate motion, and the red region exhibiting lesser motion. (E-H) The Fig shows the scatter plot among the first three principal components (PC1, PC2, and PC3) of the 66 system, with the blue region exhibiting the most pronounced motion, the white region exhibiting intermediate motion, and the red region exhibiting lesser motion. (I-L)The Fig shows the scatter plot among the first three principal components (PC1, PC2, and PC3) of the DUH system, with the blue region exhibiting the most pronounced motion, the white region exhibiting intermediate motion, and the red region exhibiting lesser motion.
(TIF)

**S9 Fig. The third time Principal component analysis plot for the complex 65 and 66 and DUH.** (A-L) The Fig shows the scatter plot among the first three principal components (PC1, PC2, and PC3) of the 65 system, with the blue region exhibiting the most pronounced motion, the white region exhibiting intermediate motion, and the red region exhibiting lesser motion. (E-H) The Fig shows the scatter plot among the first three principal components (PC1, PC2, and PC3) of the 66 system, with the blue region exhibiting the most pronounced motion, the white region exhibiting intermediate motion, and the red region exhibiting lesser motion. (I-L)The Fig shows the scatter plot among the first three principal components (PC1, PC2, and PC3) of the DUH system, with the blue region exhibiting the most pronounced motion, the white region exhibiting intermediate motion, and the red region exhibiting lesser motion.
(TIF)

**S10 Fig. The second time Dynamic interrelationship diagram of the complex.** (A) Compound 65 with DHODH (B) Compound 66 with DHODH. (C) DUH with DHODH. Positive correlations between residues are shown in cyan and negative correlations are shown in light green.
(TIF)

**S11 Fig. The third time Dynamic interrelationship diagram of the complex.** (A) Compound 65 with DHODH (B) Compound 66 with DHODH. (C) DUH with DHODH. Positive correlations between residues are shown in cyan and negative correlations are shown in light green.
(TIF)

**S12 Fig. The second time PCA-based Free-energy topography and variation of the lowest energy conformation of the protein-ligand.** (A) Compound 65 with DHODH (B) Compound 66 with DHODH. (C) DUH with DHODH.
(TIF)

**S13 Fig. The third time PCA-based Free-energy topography and variation of the lowest energy conformation of the protein-ligand.** (A) Compound 65 with DHODH (B) Compound 66 with DHODH. (C) DUH with DHODH. (TIF)

## Author contributions

**Funding acquisition:** Zhu Liang.

**Investigation:** Qu Wang.

**Methodology:** Qu Wang, Yu Hao XU, Heng Jiang, Biao Deng.

**Project administration:** Qu Wang, Yu Hao XU, Heng Jiang, Biao Deng.

**Resources:** Qu Wang, Heng Jiang, Biao Deng.

**Software:** Qu Wang, Yu Hao XU, Heng Jiang, Biao Deng.

**Supervision:** Zhu Liang.

**Writing – original draft:** Qu Wang.

**Writing – review & editing:** Qu Wang, Yu Hao XU, Biao Deng, Zhu Liang.

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
