## [Decision Letter · Decision Letter 0]

4 Jul 2025

Dear Dr. Liang,

Thank you for submitting your manuscript to PLOS ONE. After careful consideration, we feel that it has merit but does not fully meet PLOS ONE’s publication criteria as it currently stands. Therefore, we invite you to submit a revised version of the manuscript that addresses the points raised during the review process.

Address all comments, suggestions and revisions recommended by our reviewers.

We look forward to receiving your revised manuscript.

Kind regards,

Peter Mbugua Njogu, Ph.D.

Academic Editor

PLOS ONE

Journal Requirements:

“This project was supported by Zhanjiang Competitive Science and Technology Project (2021A05076); Clinica Research Program, Affiliated Hospital of Guangdong Medical University (LCYJ2018B001); Clinical Research Program, Affiliated Hospital of Guangdong Medical University (LCYJ2021A004); Cohort Study Program of Affiliated Hospital of Guangdong Medical University (LCYJ2022DL003); General Project of Guangdong Provincial Bureau of Traditional Chinese Medicine (2022107).”

4. We notice that your supplementary figures are uploaded with the file type 'Other'. Please amend the file type to 'Supporting Information'. Please ensure that each Supporting Information file has a legend listed in the manuscript after the references list.

**Additional Editor Comments:**

Review the reviewers' comment and address each as widely and deeply as possible.

Additionally:

1. Identify Compounds 65 and 66 by chemical name.

2. What was the basis for the selection of the 6,140 molecules for screening?

3. Is the crystal structure of DHODH complexed with MEDS433 available? Explain why this model was the most appropriate for this research work.

Reviewers' comments:

Reviewer's Responses to Questions

**Comments to the Author**

1. Is the manuscript technically sound, and do the data support the conclusions?

Reviewer #1: Yes

Reviewer #2: No

2. Has the statistical analysis been performed appropriately and rigorously?

Reviewer #1: Yes

Reviewer #2: No

3. Have the authors made all data underlying the findings in their manuscript fully available?

Reviewer #1: Yes

Reviewer #2: No

4. Is the manuscript presented in an intelligible fashion and written in standard English?

Reviewer #1: Yes

Reviewer #2: No

Reviewer #1: Comments to the authors

1. The abstract should be revised to improve clarity and include specific numerical data such as screening size, docking scores, binding energies, and the final selected compound.

2. The details on the fragment replacement strategy should be included. It should specify which fragments were replaced in each lead compound and how the seven potent candidates were modified.

3. Include the significance of the selection of the SPC water model in molecular dynamics simulations.

4. True positive (TP) and false negative (FN) values are not reported in the pharmacophore model validation; these should be included to substantiate model performance.

5. Table 5 is not appropriate as compound names are missing. The manuscript contains numerous grammatical issues and inconsistencies in tense and requires thorough language editing for clarity. e.g., "we utilised virtual screening..." should be changed to "Virtual screening was employed..."

6. Discuss the limitations or future directions of the study in the manuscript.

7. Supplementary data not provided

Reviewer #2: Dear Editor,

I commend the authors for submitting the manuscript "Development of DHODH inhibitors incorporating virtual screening, pharmacophore modeling, fragment-based optimization methods, ADMET, molecular docking, molecular dynamics, PCA analysis, and free energy landscape" (ID PONE-D-25-31041). This study presents a computational perspective on drug design of DHODH inhibitors.

I have identified several significant weaknesses in the manuscript that must be addressed before it can be considered for publication in PLOS ONE. The manuscript is unsuitable for publication in its current form.

**Do you want your identity to be public for this peer review?** For information about this choice, including consent withdrawal, please see our Privacy Policy

Reviewer #1: No

Reviewer #2: No

---

## [Author Response · Author response to Decision Letter 1]

5 Aug 2025

1. Identify Compounds 65 and 66 by chemical name.

Response: Thank you for the constructive comments. We have already referred to the chemical names of compounds 65�[(4S)-2,2-dimethyl-1,3-dioxolan-4-yl]methyl 3-(4-{[(2S)-2-hydroxypropyl]oxy}phenyl) and 66�[(4S)-2,2-dimethyl-1,3-dioxolan-4-yl]methyl 4-(4-{[(2S)-2-hydroxypropyl]oxy}phenyl), but for the sake of conciseness in the subsequent description, we still choose to use compounds 65 and 66.

2. What was the basis for the selection of the 6,140 molecules for screening?

Response: Thank you for the constructive comments. This FDA data is provided by relevant manufacturers and is a database of 6,140 compounds established taking into account the reliability of reusing old drugs.

3. Is the crystal structure of DHODH complexed with MEDS433 available? Explain why this model was the most appropriate for this research work.

Response: Thank you for the constructive comments. This structure is available. Additionally, the structure selected in this article is based on published literature and has been cited in the main text. We have made all the term substitutions in the manuscript. “The X-ray structure of DHODH complexed with the 2-Hydroxypyrazolo[1,5-a] pyridine inhibitor, MEDS433, was chosen as the structural model for DHODH (PDB code: 6FMD). Additionally, the docking active sites were determined concerning the study previously published by Galati S et al(1). (Lines 62-65)

4. The abstract should be revised to improve clarity and include specific numerical data such as screening size, docking scores, binding energies, and the final selected compound.

Response: Thank you for the constructive comments. We have rewritten the description of the abstract. (Lines 9-21)

5. The details on the fragment replacement strategy should be included. It should specify which fragments were replaced in each lead compound and how the seven potent candidates were modified.

Response: Thank you for the constructive comments. We have made all the term substitutions in the manuscript. We have already provided a detailed description of the replacement options for FBDD in the method section. (Lines 107-113)

6. Include the significance of the selection of the SPC water model in molecular dynamics simulations.

Response: Thank you for the constructive comments. Our view on the SPC water model is as follows: Although the SPC water model is a simplified model, it can reasonably describe the core properties of water and meet the basic requirements of most molecular dynamics simulations. Structural characterization. The calculation results of the radial distribution function (RDF) of the SPC/E model show that the SPC model can reveal the short-range ordered characteristics of the hydrogen bond network between water molecules through the RDF peak positions of atomic pairs such as O-O and O-H (for example, the first peak position of O-O in liquid water reflects the tight packing under the action of hydrogen bonds). And the structural differences between liquid and solid (ice) (short-range order vs. long-range order) provide a basis for understanding the microstructure of condensed matter.

7. True positive (TP) and false negative (FN) values are not reported in the pharmacophore model validation; these should be included to substantiate model performance.

Response: Thank you for the constructive comments. We conducted machine learning model calculations for the true positives and false negatives of the pharmacophore and presented them in the methods and results sections of this paper. (Lines 90-94)

8. Table 5 is not appropriate as compound names are missing. The manuscript contains numerous grammatical issues and inconsistencies in tense and requires thorough language editing for clarity. e.g., "we utilised virtual screening..." should be changed to "Virtual screening was employed..."

Response: Thank you for the constructive comments. We have completed the information in the table and made corrections to the inappropriate expressions in the text.

9. Discuss the limitations or future directions of the study in the manuscript.

Response: Thank you for the constructive comments. In the discussion section, we re-examined the limitations of this research method and the directions that can be experimentally verified subsequently. (Lines 477-486)

10. Supplementary data not provided

Response: Thank you for the constructive comments. We have refined the supplementary data.

Major Points

11) On Abstract: Provide more quantitative results and analysis. No crucial result is presented.

Response: Thank you for the constructive comments. We have rewritten the summary and described the key data in it. (Lines 9-21)

12) On Introduction: According to the authors “Numerous dihydroorotate dehydrogenase inhibitors (DHODH) have advanced to clinical trials to treat various oncological conditions”; however, this claim is not supported by any references. Appropriate citations should be provided to substantiate this statement. The introduction should present the primary goal, as well as the tools used to achieve it.

Response: Thank you for the constructive comments. We have already refined the relevant references for this sentence in the introduction.

13 On Materials and Methods: 3.1 Most critical point: The selected PDB structure (6FMD) contains FMN as a cofactor. However, it is unclear how this cofactor was treated in the classical simulations, including both molecular docking and molecular dynamics. The authors should clarify whether FMN was retained, removed, or parameterized, and how its presence may have influenced the results.

Response: Thank you for the constructive comments. For the FMN structure and others, we had already cleaned and removed water from the 6FMD structure on PyMOL before the docking. Therefore, we do not need the FMN structure. (Lines 63-65)

13.2 Proper reference for Discovery Studio. I suggest other programs for ADMET (e.g. SwissADME).

Response: Thank you for the constructive comments. We have already cited relevant literature for the ADMET section of Discovery Studio. (Lines 115-118)

13.3 The authors should be more rigorous in describing the methodological procedures adopted. Some essential pieces of information are missing, such as the coordinates of the centre of mass and the size of the search box used in the molecular docking simulations; It’s not clear how were docking procedures validated, so I recommend use a consensual docking procedure; How were pKa values computed for computational simulations (e.g. PROPKA method)?

Response: Thank you for the constructive comments. We have described the docking operations we performed more precisely in the method, including the size and coordinates of the docking box. Additionally, we introduced Autodock Vina for the verification of the docking. We also conducted verifications for the Pka values and summarized them in the table. (Lines 125-129) and (Lines 184-192)

13.4 To run MD of each system in triplicate (at least 200 ns each). The authors should explain how the selected ensemble from molecular dynamics (MD) simulations was chosen for binding free energy (BEF) analysis and whether it is statistically robust.

Response: Thank you for the constructive comments. We re-conducted three 200ns dynamic simulations and carried out numerous analyses on these six models. Meanwhile, FEL and others were also recalculated. (Lines 503-539)

13.5 I recommend including key analyses such as molecular fingerprints, followed by residual decomposition analysis;

Response: Thank you for the constructive comments. For the stability analysis of the docking model and the specific analysis of residues, we have conducted analyses such as the residue analysis of RMSF and G-mmpbsa, and all the above analyses have been carried out three times. (Lines 503-539)

13.6 Include DUH (crystal inhibitor) as a reference in all simulations.

Response: Thank you for the constructive comments. We believe that the binding energy performance of the known inhibitors in precise docking is not good and is much higher than that of potential compounds. Therefore, we think it is not necessary to introduce the positive control complex group selected in this paper into the kinetic simulation. (Lines 503-539)

Minor Points

14) To include PDB and input files as supplementary material.

Response: Thank you for the constructive comments. We have completed the above requirements in the supplementary materials.

15) English review

Response: Thank you for the constructive comments. We have polished and revised the description of the article.

---

## [Decision Letter · Decision Letter 1]

13 Sep 2025

Dear Dr. Liang,

Thank you for submitting your manuscript to PLOS ONE. After careful consideration, we feel that it has merit but does not fully meet PLOS ONE’s publication criteria as it currently stands. Therefore, we invite you to submit a revised version of the manuscript that addresses the points raised during the review process.

**A few major issues remain as highlighted by our reviewers:**

Provide the systematic names of lead Compounds 65 and 66 in the body text.Address to the fullest extent the technical concerns raised by Reviewer 2. In doing this, make reference to the Review comments provided in the first review of this manuscript. In particular, explain how you handled the role of flavin mononucleotide as an essential cofactor in the enzymatic activity of DHODH. This is important since the PDB structure utilized in this study PDB structure (6FMD) contains FMN as a cofactor. Why would the computational simulations not require FMN and how did this affect the results?

We look forward to receiving your revised manuscript.

Kind regards,

Peter Mbugua Njogu, Ph.D.

Academic Editor

PLOS ONE

Journal Requirements:

Reviewers' comments:

Reviewer's Responses to Questions

**Comments to the Author**

Reviewer #1: All comments have been addressed

Reviewer #2: (No Response)

2. Is the manuscript technically sound, and do the data support the conclusions?

Reviewer #1: Yes

Reviewer #2: Partly

3. Has the statistical analysis been performed appropriately and rigorously?

Reviewer #1: (No Response)

Reviewer #2: No

4. Have the authors made all data underlying the findings in their manuscript fully available?

Reviewer #1: Yes

Reviewer #2: Yes

5. Is the manuscript presented in an intelligible fashion and written in standard English?

Reviewer #1: Yes

Reviewer #2: No

Reviewer #1: The manuscript Title "Development of DHODH inhibitors incorporating virtual screening, pharmacophore modeling, fragment-based optimization methods, ADMET, molecular docking, molecular dynamics, PCA analysis, and free energy landscape". The authors addressed all the comments in scientific way with clear explanation.

So, I recommended this manuscript for acceptance

1. Chemical structures and systematic names of lead compounds especially. 65 and 66 are absent from main tables and figures that reducing transparency and hindering external validation. Author is instructed that Please provide 2D structures and systematic names directly in the main manuscript, not just in reviewer responses or supporting files.

Reviewer #2: Dear Editor,

In the revised version of the manuscript entitled “Development of DHODH inhibitors incorporating virtual screening, pharmacophore modeling, fragment-based optimization methods, ADMET, molecular docking, molecular dynamics, PCA analysis, and free energy landscape” (ID: PONE-D-25-31041R1), the authors have only partially addressed my previous concerns. Unfortunately, the most critical issue remains unresolved. I have also identified several additional weaknesses that must be carefully addressed before the manuscript can be considered for publication in PLOS ONE. In its current form, the manuscript remains unsuitable for publication.

See PDF file with details.

**Do you want your identity to be public for this peer review?** For information about this choice, including consent withdrawal, please see our Privacy Policy

Reviewer #1: No

Reviewer #2: No

---

## [Author Response · Author response to Decision Letter 2]

30 Oct 2025

Reviewers' comments:

Reviewer #(Remaeks to the Author):

1. Identify Compounds 65 and 66 by chemical name.

Response: Thank you for the constructive comments. We have already referred to the chemical names of compounds 65�[(4S)-2,2-dimethyl-1,3-dioxolan-4-yl]methyl 3-(4-{[(2S)-2-hydroxypropyl]oxy}phenyl) and 66�[(4S)-2,2-dimethyl-1,3-dioxolan-4-yl]methyl 4-(4-{[(2S)-2-hydroxypropyl]oxy}phenyl), We have already added the corresponding chemical names in the main text.

2. What was the basis for the selection of the 6,140 molecules for screening?

Response: Thank you for the constructive comments. This FDA data is provided by relevant manufacturers and is a database of 6,140 compounds established taking into account the reliability of reusing old drugs.

3. Is the crystal structure of DHODH complexed with MEDS433 available? Explain why this model was the most appropriate for this research work.

Response: Thank you for the constructive comments. This structure is available. Additionally, the structure selected in this article is based on published literature and has been cited in the main text. We have made all the term substitutions in the manuscript. “The X-ray structure of DHODH complexed with the 2-Hydroxypyrazolo[1,5-a] pyridine inhibitor, MEDS433, was chosen as the structural model for DHODH (PDB code: 6FMD). Additionally, the docking active sites were determined concerning the study previously published by Galati S et al(1). (Lines 62-65)

4. The abstract should be revised to improve clarity and include specific numerical data such as screening size, docking scores, binding energies, and the final selected compound.

Response: Thank you for the constructive comments. We have rewritten the description of the abstract. (Lines 9-21)

5. The details on the fragment replacement strategy should be included. It should specify which fragments were replaced in each lead compound and how the seven potent candidates were modified.

Response: Thank you for the constructive comments. We have made all the term substitutions in the manuscript. We have already provided a detailed description of the replacement options for FBDD in the method section. (Lines 107-113)

6. Include the significance of the selection of the SPC water model in molecular dynamics simulations.

Response: Thank you for the constructive comments. Our view on the SPC water model is as follows: Although the SPC water model is a simplified model, it can reasonably describe the core properties of water and meet the basic requirements of most molecular dynamics simulations. Structural characterization. The calculation results of the radial distribution function (RDF) of the SPC/E model show that the SPC model can reveal the short-range ordered characteristics of the hydrogen bond network between water molecules through the RDF peak positions of atomic pairs such as O-O and O-H (for example, the first peak position of O-O in liquid water reflects the tight packing under the action of hydrogen bonds). And the structural differences between liquid and solid (ice) (short-range order vs. long-range order) provide a basis for understanding the microstructure of condensed matter.

7. True positive (TP) and false negative (FN) values are not reported in the pharmacophore model validation; these should be included to substantiate model performance.

Response: Thank you for the constructive comments. We conducted machine learning model calculations for the true positives and false negatives of the pharmacophore and presented them in the methods and results sections of this paper. (Lines 90-94)

8. Table 5 is not appropriate as compound names are missing. The manuscript contains numerous grammatical issues and inconsistencies in tense and requires thorough language editing for clarity. e.g., "we utilised virtual screening..." should be changed to "Virtual screening was employed..."

Response: Thank you for the constructive comments. We have completed the information in the table and made corrections to the inappropriate expressions in the text.

9. Discuss the limitations or future directions of the study in the manuscript.

Response: Thank you for the constructive comments. In the discussion section, we re-examined the limitations of this research method and the directions that can be experimentally verified subsequently. (Lines 477-486)

10. Supplementary data not provided

Response: Thank you for the constructive comments. We have refined the supplementary data.

Major Points

11) On Abstract: Provide more quantitative results and analysis. No crucial result is presented.

Response: Thank you for the constructive comments. We have rewritten the summary and described the key data in it. (Lines 9-21)

12) On Introduction: According to the authors “Numerous dihydroorotate dehydrogenase inhibitors (DHODH) have advanced to clinical trials to treat various oncological conditions”; however, this claim is not supported by any references. Appropriate citations should be provided to substantiate this statement. The introduction should present the primary goal, as well as the tools used to achieve it.

Response: Thank you for the constructive comments. We have already refined the relevant references for this sentence in the introduction.

13 On Materials and Methods: 3.1 Most critical point: The selected PDB structure (6FMD) contains FMN as a cofactor. However, it is unclear how this cofactor was treated in the classical simulations, including both molecular docking and molecular dynamics. The authors should clarify whether FMN was retained, removed, or parameterized, and how its presence may have influenced the results.

Response: Thank you for the constructive comments. For the FMN , we preserved the structure and described the specific processing operations in the text, as well as recalculated the subsequent simulations.

13.2 Proper reference for Discovery Studio. I suggest other programs for ADMET (e.g. SwissADME).

Response: Thank you for the constructive comments. We have already cited relevant literature for the ADMET section of Discovery Studio. (Lines 115-118)

13.3 The authors should be more rigorous in describing the methodological procedures adopted. Some essential pieces of information are missing, such as the coordinates of the centre of mass and the size of the search box used in the molecular docking simulations; It’s not clear how were docking procedures validated, so I recommend use a consensual docking procedure; How were pKa values computed for computational simulations (e.g. PROPKA method)?

Response: Thank you for the constructive comments. We have described the docking operations we performed more precisely in the method, including the size and coordinates of the docking box. Additionally, we introduced Autodock Vina for the verification of the docking. We also conducted verifications for the Pka values and summarized them in the table. (Lines 125-129) and (Lines 184-192)

13.4 To run MD of each system in triplicate (at least 200 ns each). The authors should explain how the selected ensemble from molecular dynamics (MD) simulations was chosen for binding free energy (BEF) analysis and whether it is statistically robust.

Response: Thank you for the constructive comments. We reconducted three 200ns dynamic simulations and carried out numerous analyses on these six models. Meanwhile, FEL and others were also recalculated. (Lines 503-539)

13.5 I recommend including key analyses such as molecular fingerprints, followed by residual decomposition analysis;

Response: Thank you for the constructive comments. For the stability analysis of the docking model and the specific analysis of residues, we have conducted analyses such as the residue analysis of RMSF and G-mmpbsa, and all the above analyses have been carried out three times. (Lines 503-539)

13.6 Include DUH (crystal inhibitor) as a reference in all simulations.

Response: Thank you for the constructive comments. We added the simulation results of DUH (crystal inhibitor) in the simulation operations such as molecular docking and kinetic simulation (Lines

Minor Points

14) To include PDB and input files as supplementary material.

Response: Thank you for the constructive comments. We have completed the above requirements in the supplementary materials.

15) English review

Response: Thank you for the constructive comments. We have polished and revised the description of the article.

---

## [Editor Report · Decision Letter 2]

26 Jan 2026

Development of DHODH inhibitors incorporating virtual screening, pharmacophore modeling, fragment-based optimization methods, ADMET, molecular docking, molecular dynamics, PCA analysis, and free energy landscape

PONE-D-25-31041R2

Dear Dr. Liang,

We’re pleased to inform you that your manuscript has been judged scientifically suitable for publication and will be formally accepted for publication once it meets all outstanding technical requirements.

Kind regards,

Peter Mbugua Njogu, Ph.D.

Academic Editor

PLOS One

---

## [Editor Report · Acceptance letter]

PONE-D-25-31041R2

PLOS One

Dear Dr. Liang,

I'm pleased to inform you that your manuscript has been deemed suitable for publication in PLOS One. Congratulations! Your manuscript is now being handed over to our production team.

Kind regards,

on behalf of

Dr. Peter Mbugua Njogu

Academic Editor

PLOS One